# Investigating the synergistic potential Si and biochar to immobilize Ni in a Ni-contaminated calcareous soil after *Zea mays* L. cultivation

Hamid Reza Boostani[1], Ailsa G. Hardie[2], Mahdi Najafi-Ghiri[1], Ehsan Bijanzadeh[3], Dariush Khalili[4], Esmaeil Farrokhnejad[1]

[1]Department of Soil Science, College of agriculture and natural resources of Darab, Shiraz University, Darab74591, Iran.
[2]Department of Soil Science, Faculty of AgriSciences, Stellenbosch University, Private Bag X1, Matieland 7602, South Africa
[3]Department of agroecology, College of agriculture and natural resources of Darab, Shiraz University, Darab 74591, Iran
[4]Department of Chemistry, College of Sciences, Shiraz University, Shiraz 71454, Iran

*Correspondence to*: Hamid Reza Boostani (hr.boostani@shirazu.ac.ir)

**Abstract.** In Iran, a significant percentage of agricultural soils are contaminated with a range of potentially toxic elements (PTEs), including Ni, which need to be remediated to prevent their entry into the food chain**.** Silicon (Si) is a beneficial plant element that has been shown to mitigate the effects of PTEs on crops. Biochar is a soil amendment that sequesters soil carbon, and that can immobilize PTEs and enhance crop growth in soils. No previous studies have examined the potentially synergistic effect of Si and biochar on Ni concentration in soil chemical fractions and immobilization thereof. Therefore, the aim of this study was to examine the interactive effects of Si and biochar, to reduce Ni bioavailability and its corresponding uptake in corn (*Zea Mays*) in a calcareous soil. A 90-day factorial greenhouse study with corn was conducted. Si application levels were applied at 0 ($S_0$), 250 ($S_1$) and 500 ($S_2$) mg Si $kg^{-1}$ soil and biochar treatments (3% wt.) including rice husk (RH) and sheep manure (SM) biochars produced at $300°C$ and $500°C$ (SM300, SM500, RH300 and RH500). At harvest, Ni concentration in corn shoots, Ni content in soil chemical fractions and release kinetic of DPTA-extractable Ni were determined. Simultaneous utilization of Si and SM biochars led to a synergistic reduction (15-36%) of Ni content in the soluble and exchangeable fractions compared to application of Si (5-9%) and SM (5-7%) biochars separately. The application of the Si and biochars also decreased DPTA-extractable Ni and Ni content in corn shoots (by up to 57%), with the combined application of SM500+$S_2$ being the most effective. These effects were attributed to the transfer of Ni in soil from more bioavailable fractions to more stable iron oxide bound fractions, related to soil pH increase. The SM500 was likely the most effective biochar due to its higher alkalinity and lower acidic functional group content which enhanced Ni sorption reactions with Si. The study demonstrates the synergistic potential of Si and sheep manure biochar at immobilizing Ni in contaminated calcareous soils.

## 1 Introduction

One of the most important ways for potentially toxic elements (PTEs) to enter the human food chain is through the consumption of plants grown in soils contaminated with PTEs. Potentially toxic elements pollute soil environments as a result of mining, metal smelting, use of sewage sludge and domestic and industrial effluents in agriculture especially in developing countries (Liu et al., 2018). Potentially toxic elements in soils cannot undergo biodegradation by

living organisms, so they possess great stability and longevity in the soil (Poznanović Spahić et
al., 2019). Unlike other PTEs found in soils, such as mercury (Hg), cadmium (Cd) and lead (Pb),
nickel (Ni) is essential for plant growth at very low concentrations. Nevertheless, at elevated
contents (>35 mg Ni kg$^{-1}$ soil), Ni causes many physiological and morphological malfunctions in
plants and severely stunts their growth (Shahzad et al., 2018; Antoniadis et al., 2017). In a study
conducted by Shahbazi et al. (2022), the Ni weighted average concentration of the cultivated lands
of Iran in the vicinity of the industrial areas was reported 350 mg kg$^{-1}$ soil. In these soils, the
pollution index (the ratio of the element concentration to the standard concentration) calculated
for the Ni is greater than 5, which indicates a severe degree of pollution from the point of view of
environmental protection. Shahbazi et al. (2020) collected 711 agricultural soil samples located at
different climate zones (extra arid, arid, semi-arid, Mediterranean, semi humid, humid and per-
humid based on the de Martonne classification system) of Iran and reported that the Ni content in
the soils was between 2.79 mg kg$^{-1}$ and 770 mg kg$^{-1}$ with an average of 68 mg kg$^{-1}$ soil. The results
showed that the concentration of Ni in 11.3% of these soils was higher than the threshold value.
Removing PTEs from contaminated sites via traditional methods such as pump and treat
technologies, soil washing and excavation is very expensive and time-consuming, therefore, for
plant cultivation in these areas, low-cost and effective methods should be sought to stabilize PTEs
in soils and prevent them from being transferred to the plant (Gao et al., 2023).
Silicon (Si) is a valuable nutrient for plant growth, and it is only considered essential for
some plant species such as rice. Applying Si to the soil can enhance plant resistance against
biological and non-biological tensions, including physiological stress caused by PTEs in soil (Bhat
et al., 2019; Yan et al., 2018). The use of Si to promote plant growth and mitigate the toxicity of
PTEs is becoming increasingly popular in agriculture (Li, 2019; Adrees et al., 2015). The
application of Si in soils contaminated with PTEs may reduce the bioavailability of PTEs by
increasing soil pH, increasing the secretion of organic ligands by the roots and the formation of
insoluble compounds with PTEs, and ultimately enhancing plant growth (Bhat et al., 2019; Xiao
et al., 2021). The soil pH increase associated with Si application is attributed to the hydrolysis
reaction of the silicate anion in soil solution which generates hydroxyl ions (Ma et al. 2021).
Biochar can be used for a number of applications including as a soil amendment that
sequesters soil carbon (C) and for stabilization of PTEs in polluted sites (El-Naggar et al., 2018).
Biochar is a carbon-rich, porous organic material which is prepared in a limited or no oxygen
conditions by pyrolysis of organic wastes, including crop and animal residues, urban waste, wood
byproduct (Vickers, 2017; Ankita Rao et al., 2023). The organic surface functional groups of
biochar such as carboxylic and phenolic groups provide cation exchange capacity in soils
(Tomczyk et al., 2020). Addition of biochar to the soil not only improves the soil chemical and
physical properties, but also reduces the bioavailability of PTEs in contaminated soils through
some physicochemical processes such as sedimentation, complexation, and electrostatic
adsorption (Bandara et al., 2020; Deng et al., 2019; Derakhshan Nejad et al., 2018). The
complexation of Ni with oxygen-containing functional groups on biochar surfaces including
carboxyl, ether, carbonyl, and hydroxyl, has been identified as a key mechanism for Ni
immobilization in soil (Alam et al., 2018; El-Naggar et al., 2018). Electrostatic attraction of Ni by
negatively charged functional groups on the surfaces of biochar is another potential mechanism
for Ni stabilization in soil (Ahmad et al., 2014). Increased soil pH following the application of
biochar also promotes Ni adsorption reactions (Uchimiya et al., 2010). However, the efficiency of
biochar prepared from different feedstocks and under different production conditions in stabilizing
PTEs in soils can vary significantly (Dey et al., 2023).
Potentially toxic elements in soil can exist in different chemical fractions such as water
soluble and exchangeable (WsEx), bound to carbonates (Car), organic matter (OM), iron and
manganese oxides (FeMnOx) and residual (Res) (found in minerals) (Singh et al., 1988). The
bioavailability of these forms differs, the WsEx fraction has the highest bioavailability and the Res
form is considered unusable by plants. PTEs in other chemical fractions in soils could be
potentially accessible for plant roots depending on soil characteristics such as soil texture, soil pH
and soil organic matter content (Kamali et al., 2011; Bharti et al., 2018). The diethylene triamine
penta-acetic acid (DTPA) extraction is commonly employed for assessing Ni availability in
calcareous soils (Lindsay and Norvell, 1978). However, it is important to acknowledge that this
methodology solely assesses Ni availability for plants, while the quantity of released Ni may vary
across distinct stages of plant development. Consequently, the examination of alterations in
extractable Ni levels over time using the DTPA solution can prove valuable in estimating
bioavailability of Ni in soil. The PTEs desorption capacities of soils are anticipated to be contingent
upon factors such as soil pH, cation exchange capacity, the specific nature of metal ions, and the
source of the metals (Kandpal et al., 2005). Furthermore, the release kinetic parameters can provide
insight into the bonding mechanisms of PTEs in soils and their potential risk for leaching into
groundwater or surface water (El-Naggar et al., 2021). Therefore, sequential extraction methods
and release kinetic models have been employed to assess the efficacy of amendment materials in
stabilizing PTEs in contaminated soils. Xiao et al. (2021) found that addition of mineral Si fertilizer
to a contaminated paddy soil caused a significant decrease in the Cd and Pb fractions bound to
carbonates and iron-manganese oxides while the residual and organic matter-bound forms
experienced a notable increase. In another study, application of cotton residue biochar (1.5 wt. %)
to a calcareous soil with a sandy loam textural class containing different levels of Cd contamination
was more efficacious than corn and wheat straw biochars in decreasing Cd content in the WsEx
and Car fractions and enhancing Cd concentration in the Res fraction. In addition, application of
cotton residue biochar decreased EDTA-extractable Cd by 45–52% compared to the control
(Boostani et al., 2023b).
As both biochars and Si are economical and effective soil amendments to reduce plant
uptake of PTEs and stress in contaminated soils, their potential synergistic effect on the
immobilization of PTEs in soils should be further investigated. Currently, no previous studies have
examined the combined application effects of Si and biochars on the Ni content in various soil
chemical fractions and release kinetic of Ni in calcareous soils. The primary objective of the
present study was to elucidate the interaction of biochars and Si levels, to reduce bioavailability of
Ni in soil and its corresponding accumulation in corn (*Zea Mays* L. 604) plant. Additionally, the
study sought to elucidate the underlying soil chemical mechanisms that are likely to be responsible
for such effects.

## 2 Materials and methods


2.1 Soil sampling, characterization and Ni treatment
A composite soil sample from the surface layer (0-30 cm) was collected with an auger at
the research farm of the College of Agriculture and Natural Resources in Darab, southern Iran (28
$^{\circ}$ 45$'$ 0.99$''$ N 54$^{\circ}$ 26$'$ 52.14$''$ E, Elevation 1105 m) (Fig. 1). The climate, mean annual
precipitation, soil moisture and thermal regimes of the studied area were semi-arid, 250 mm, ustic

and hyperthermic respectively. The composite soil sample was placed in polyethylene bags in the field and then transported to laboratory where it was immediately air-dried, passed through a 2 mm sieve and then stored at room temperature until the physicochemical analysis was performed. Soil sand, silt and clay content were determined by the hydrometer method (Gee and Bauder, 1986). Soil pH and EC were determined using a saturated paste (Rhoades, 1996), while organic matter was determined using Walkley-Black procedure (Nelson and Sommers, 1996). Calcium carbonate equivalent (CCE) was determined by acid neutralization (Loeppert and Suarez, 1996), while cation exchange capacity was determined using 1M ammonium acetate (Merck, 99%) method (Sumner and Miller, 1996). Available Ni was determined using DPTA (Merck, 99%) extraction (Lindsay and Norvell, 1978). Plastic containers were filled with two kilograms of soil and then 500 ml $NiCl_2$ (Merck, 99%) solution was mixed into to them to achieve a Ni concentration of 300 mg Ni $kg^{-1}$ soil. The Ni-treated soil samples were then allowed to dry out at room temperature, and then rewetted to field capacity using deionized water and allowed to dry out again. The rewetting and room temperature drying cycle was repeated three times to allow the Ni to equilibrate with the soil. The repeated wetting and drying cycles were performed to simulate field processes (Boostani et al., 2023c). The incubation period was 60 days and the ambient temperature was 25±2 ℃.

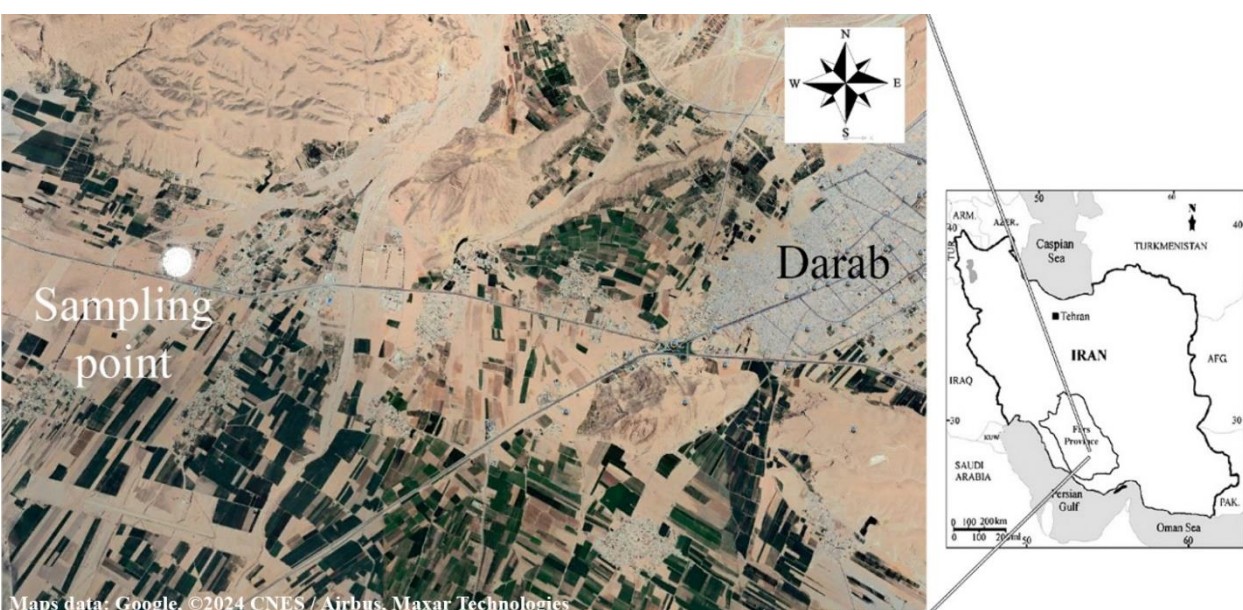

**Fig. 1.** The location of soil sampling in Darab region, southern Iran.

## 2.2 Production of biochar and its properties

The sheep manure and rice husk were respectively procured from an active animal husbandry and rice mill factory situated in the Darab region, Fars province, Iran. Subsequently, the raw materials underwent a 1-week period of air-drying, followed by electrical milling and sieving through a 2 mm mesh. A slow pyrolysis procedure (2 h at 300 °C and 500 °C) in an oxygen-limited environment was carried out to generate biochars from feedstocks (Anand et al., 2023).

The generated biochars were then cooled at ambient temperature and sieved with a 0.5 mm mesh to ensure consistent particle size. The chemical characteristics of the biochars were assessed using the following standard methods. Biochar pH and EC were determined in a 1:10 deionized water suspension (Sun et al., 2014), while CEC was determined using the method of Abdelhafez et al. (2014). Biochar total C, N and H contents were determined by elemental analyzer (ThermoFinnigan Flash EA 1112 Series, Thermoscientific, USA). Biochar moisture and ash content were determined by heating in an oven, while the O+S content was calculated by subtraction of C, N, H, ash and moisture content from total biochar mass (Keiluweit et al., 2010). Biochar total Ni content was determined by combustion and dissolution of the ash in 2M HCl (Merck, 37%) (Boostani et al., 2018b). The Ni content in the acid solution was determined using atomic absorption spectroscopy (AAS) (PG 990, PG Instruments Ltd., UK). Surface functional groups of the biochars were assessed using Fourier Transform Infrared (FTIR) spectroscopy using a Shimadzu DR-8001 instrument and KBr pellet transmission method. The morphology of surface biochar was assessed using scanning electron microscopy (SEM) (TESCAN-Vega3, Czech Republic).

2.3 Greenhouse experiment

A completely randomized factorial experiment was conducted in a greenhouse environment with three replications. The first factor consisted of the biochar treatments including rice husk (RH) and sheep manure (SM) generated at 300 °C and 500 °C (Control (C) (with no biochar), SM300, SM500, RH300 and RH500), each at the level of 3%wt. The second factor included Si application levels (0 ($S_0$), 250 ($S_1$) and 500 ($S_2$) mg Si $kg^{-1}$ soil) supplied as $Na_2SiO_3$ (Sigma Aldrich, 98%) solution. Based on the experimental design, Si levels were added to the 2 kg of Ni-treated soil samples and after drying the soil and mixing it, the prepared biochars were added to the required amount. Immediately after that, the treated soil samples were transferred to plastic pots (45 pieces each containing 2 kg soil) and to facilitate the required reactions, the moisture content of the samples was kept at field capacity level for a duration of two weeks. Thereafter, 6 corn seeds (*Zea mays* L. 604) were planted in each pot, and at the 4-leaf stage, 2 plants were kept in each pot until the end of cultivation. During the growth of the plant, distilled water was used to maintain the soil moisture content in the pots at field capacity. After 90 days, the plants were harvested at the soil interface, rinsed with distilled water to remove contamination, immediately air-dried and kept for Ni determination of plant shoots. After separating the roots, the soil from the pots was immediately air-dried, and then passed through a 2 mm sieve and stored in labelled polyethylene bags at room temperature, to be subsequently utilized for performing Ni sequential extraction and release kinetics.

2.4 Sequential extraction procedure

The present study employed a successive extraction technique (Singh et al., 1988) to fractionate Ni in the following soil chemical fractions, namely water-soluble and exchangeable (WsEx), carbonate-bound (Car), organic matter-bound (OM), manganese oxide-bound (MnOx), amorphous iron oxide-bound (AFeOx), crystalline iron oxide-bound (CFeOx), and residual (Res). The methodological specifics are provided in Table 1.

**Table 1**
Successive extraction technique of Singh et al. (1988)

| Chemical speciation containing Ni | acronym | Duration of agitation (h) | Extractants | Relative density (g.cm$^{-3}$) |
|---|---|---|---|---|
| Exchangeable and soluble | WsEx | 2.0 | 1 M magnesium nitrate (Merck, 98%) | 1.10 |
| Carbonate | Car | 5.0 | 1 M sodium acetate (Merck, 99%) (pH=5) | 1.04 |
| Organic | OM | 0.5 | 0.7 M sodium hypocholoride (pH=8.5) | 1.00 |
| Mn oxide | MnOx | 0.5 | 0.1 M hydroxyl amine hydrochloride (Merck, 98%) (pH=2 by nitric acid (Merck, 65%)) | 1.00 |
| Amorphous Fe oxides | AFeOx | 0.5 | 0.25 M hydroxyl amine hydrochloride (Merck, 98%) + 0.25 M choloridric acid (Merck, 37%) | 1.01 |
| Crystalline Fe oxides | CFeOx | 0.5 | 0.2 M ammonium oxalate (Merck, 99%) + 0.2 M oxalic acid (Merck, 99%) + 0.1 M ascorbic acid (Merck, 99.7%) | 1.02 |

201

## 2.5 Release kinetics experiment

A 50 ml centrifuge tube was filled with 10 g of soil. After that, 20 ml of DTPA solution (0.005 M DTPA (Merck, 99%) + 0.1 M tri-ethanol amine (Merck, 99%) + 0.01 M calcium chloride (Merck, 97%)) (pH: 7.3) (Lindsay and Norvell, 1978) was added to the soil. The soil-DTPA mixtures were stirred for specific periods of time, i.e. 5, 15, 30, 60, 120, 360, 720 and 1440 minutes at a constant temperature (25 $\pm$2 °C). After each stirring time, the soil suspension was centrifuged (2683 $\times$ g) to separate the soil particles from the liquid phase. Atomic absorption spectroscopy (AAS) (PG 990, PG Instruments Ltd., UK) was used to analyze the Ni concentration in the liquid phase. The Ni concentration in the liquid phase versus time was plotted to obtain a Ni release kinetic curve. A total of seven kinetic models namely order models (zero, first, second and third), parabolic diffusion, power function and simple Elovich were assessed to fit the Ni release data. The best models for describing the data were selected according to the maximum value of the coefficient of determination ($R^2$) and the minimum amount of the standard error of estimate (SEE) (Nasrabadi et al., 2022).

## 2.6 Data analysis

The ANOVA test was utilized to assess treatments effects between the individual and combined biochar and silicon treatments. Additionally, a comparison of means was conducted using the MSTATC computer program, applying Duncan's test with a significance level of 5%. Figures were generated using Excel 2013 software. Pearson correlation coefficients among parameters in the dataset were determined using SPSS 12.0.

## 3 Results and Discussion

### 3.1 Soil characteristics

The soil used in the study prior to experimental treatment, exhibited a sandy loam texture and possessed alkaline properties with significant calcium carbonate content (calcareous soil), while not being classified as saline (Table 2). The quantity of soil organic matter (OM) and cation exchange capacity (CEC) were extremely low (Table 2) compared to the values (OM: 1.20-7.10%,

CEC: 19.5-55.9 cmol$_{(+)}$ kg$^{-1}$) documented by Rassaei et al. (2020) for calcareous soils of Fars
province, southern Iran. The soils in Iran mainly originate from calcareous alluvium under xeric,
ustic or aridic and mesic, thermic or hyperthermic moisture and temperature regimes, respectively.
These soils have varied properties such as calcium carbonate equivalent (1-81%), clay content (1-
75%), EC (0.40-49.0 dS m$^{-1}$), organic matter (0.10-21.5%) and gypsum content (0-91%) (Ghiri et
al., 2011). Furthermore, it should be noted that the concentration of available Ni extractable by
DTPA in the soil was low (Table 2) compared to the mean value (0.60 mg Ni kg$^{-1}$) reported by
Jalali et al. (2022) for calcareous soils located at western Iran.

**Table 2**
Certain physicochemical attributes of the soil prior to cultivation.

| | |
|---|---|
| Sand (%) | 58.0 |
| Silt (%) | 30.0 |
| Clay (%) | 12.0 |
| Soil textural class | Sandy loam |
| pH$_{(s)}$ | 7.59 |
| EC (dS m$^{-1}$) | 2.60 |
| CCE (%) | 55.0 |
| OM (%) | 0.50 |
| CEC (cmol$_{(+)}$kg$^{-1}$) | 11.7 |
| Total Ni (mg kg$^{-1}$) | 28.0 |
| Ni-DTPA (mg kg$^{-1}$) | 0.39 |

Notes: EC, electrical conductivity; OM, organic matter; CCE, calcium carbonate equivalent; CEC, cation exchange capacity.


3.2 Chemical characteristics of the biochars
As the pyrolysis temperature rose from 300 °C to 500 °C, the SM biochars demonstrated
elevated pH and EC values, with the highest levels observed at the highest temperature (Table 3).
The elevated levels of alkali salts, which are reflected in the high ash content (Table 3), are the
contributing factor behind this observation in the SM biochars in comparison to the RH biochars.
Plant-based biochars commonly exhibit reduced levels of dissolved solids in comparison to
animal-based biochars (Sun et al., 2014). The SM300 biochar possessed the highest CEC value of
19.70 cmol$_{(+)}$ kg$^{-1}$. The observed phenomenon may be attributed to the diminution of surface
functional groups, namely carboxyl and phenol, at elevated pyrolysis temperatures. These groups
are predominantly responsible for facilitating the cation exchange capacity (CEC) of biochars
(Tomczyk et al., 2020). As the pyrolysis temperature increased, there was an observed increase in
the C content of the biochars, and a corresponding decrease in the content of hydrogen, oxygen,
and nitrogen (Table 3). The observed increase in the concentration of C as pyrolysis temperature
rises is consistent with a concomitant rise in the degree of carbonization. The observed reduction
in the levels of H and O might be attributed to the occurrence of dehydration reactions,
decomposition of oxygenated bonds, and the liberation of low molecular weight byproducts rich
in H and O, as recently noted by Zhao et al. (2017). Nitrogen compound volatilization explains the
diminished N content of the biochars at elevated pyrolysis temperatures. The ratios of H:C and
O:C are significant indicators of the aromaticity and polarity of biochars; the lower the ratios the

more condensed aromatic C the biochar contains (Chatterjee et al., 2020). The results shown in Table 3 indicate that the H:C and O:C mole ratios showed a gradual decrease as the pyrolysis temperature was increased, which can be interpreted as a sign of improved carbonization of the biochars (Zhao et al., 2017). The Ni content in the biochars derived from rice husk was below detection, whereas a limited quantity of Ni was detected in the biochars produced from sheep manure (Table 3).

**Table 3**
Some physical and chemical properties of the biochars.

|  | SM300 | SM500 | RH300 | RH500 |
|---|---|---|---|---|
| pH (1:20) | 9.96 | 11.0 | 9.00 | 10.3 |
| EC (1:20) (dS m$^{-1}$) | 3.94 | 4.28 | 0.84 | 1.17 |
| CEC (cmol$_+$ kg$^{-1}$) | 19.7 | 18.9 | 18.9 | 15.3 |
| C (%) | 25.4 | 31.8 | 45.0 | 50.0 |
| H (%) | 1.85 | 0.80 | 2.28 | 1.06 |
| N (%) | 2.10 | 1.57 | 1.30 | 1.10 |
| Ni (mg kg$^{-1}$) | 3.00 | 15.4 | Nd | Nd |
| Moisture content (%) | 1.91 | 1.82 | 2.65 | 2.37 |
| Ash content (%) | 53.8 | 60.0 | 34.2 | 44.8 |
| H:C mole ratio | 0.87 | 0.30 | 0.60 | 0.25 |
| O+S:C mole ratio | 0.44 | 0.09 | 0.24 | 0.01 |

Notes: SM300, sheep manure biochar generated at 300 °C; SM500, sheep manure biochar generated at 500 °C; RH300, rice husk biochar produced at 300 °C; RH500, rice husk biochar produced at 500 °C; CEC, cation exchange capacity; EC, electrical conductivity; Nd, non-detectable.

## 3.3 Biochar analysis using FTIR and SEM

The FTIR spectra of the SM and RH biochars are shown in Figure 2. The SM and RH biochars produced at 300 °C contained a higher content of carboxyl groups (1700 cm$^{-1}$) (Keiluweit et al., 2010) than the biochars produced at 500 °C, which is in agreement with the O:C values of the biochars (Table 3). All of the biochars contained absorption bands associated with lignin (1430 cm$^{-1}$) and cellulose (1030 - 1160 cm$^{-1}$) (Keiluweit et al., 2010). The SM biochar contained more calcite than the RH biochar as indicated by the greater intensity of calcite characteristic peaks at 1432, 875, and 711cm$^{-1}$ (Myszka et al., 2019) in the SM biochars (Fig. 2a). There was also evidence of the presence of Ca oxalate in the SM biochars, indicated by the characteristic peaks at 1618, 780 and 518 cm$^{-1}$ (Maruyama et al., 2023). All the biochars contained silica as indicated by the intense silica absorption peaks at 1100, 800 and 470 cm$^{-1}$ (Zemnukhova et al., 2015).

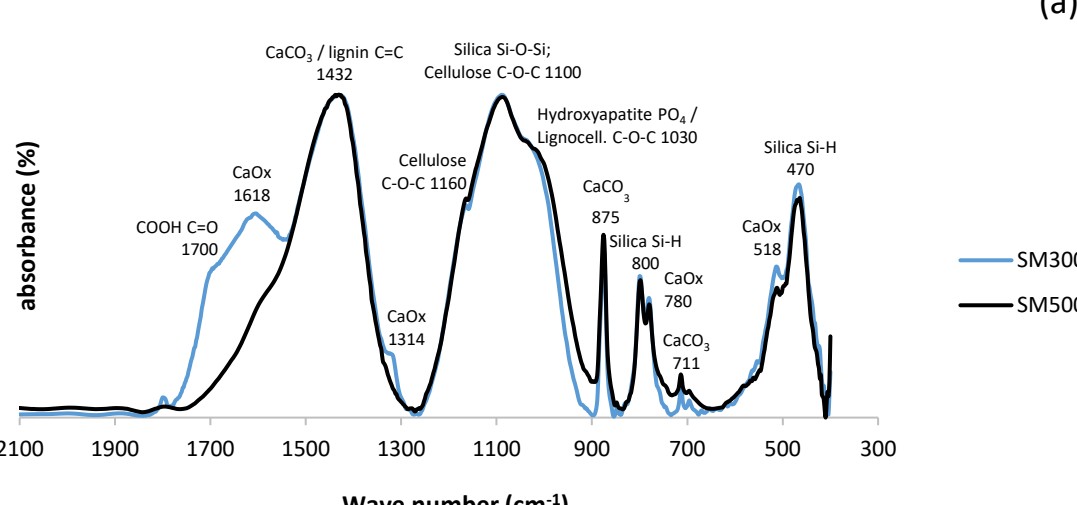

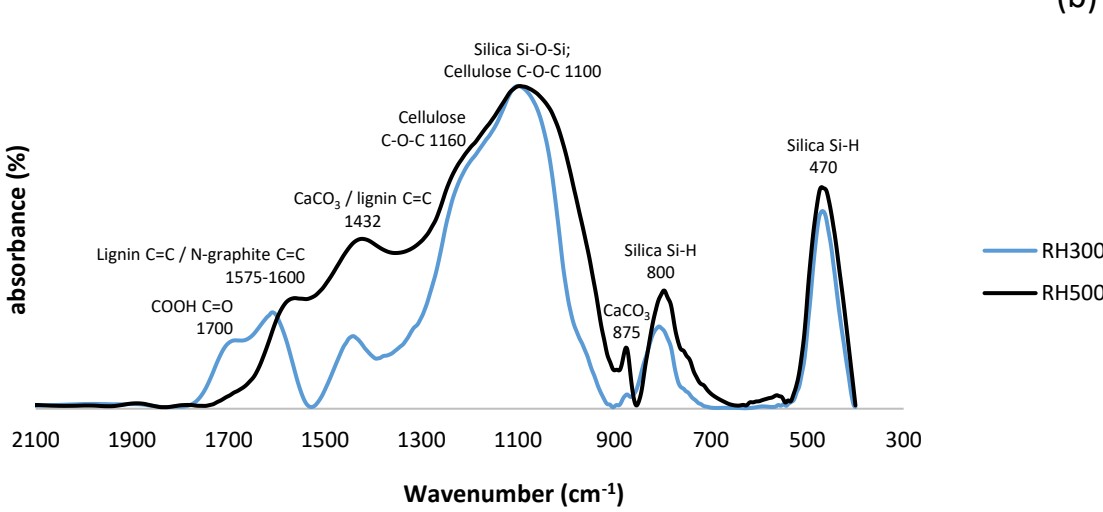

**Fig. 2.** FTIR spectra the biochars in the wave number range of 400-2000 cm[-1]. Notes: SM300, sheep manure biochar produced at 300 °C; SM500, sheep manure biochar produced at 500 °C; RH300, rice husk biochar produced at 300 °C; RH500, rice husk biochar produced at 500 °C.

The SEM images of the SM and RH biochars are shown in Figure 3. The morphology of the biochars became more rigid and porous at higher temperatures, as evidenced by the cell wall shrinkage attributed to devolatilization of organic tissues (Claoston et al., 2014).

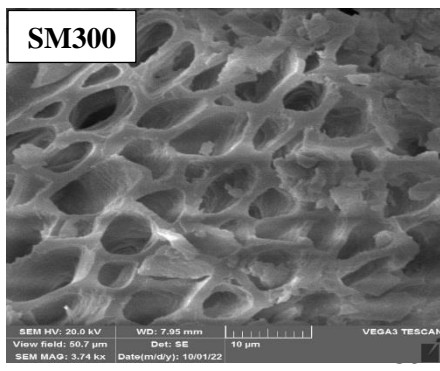
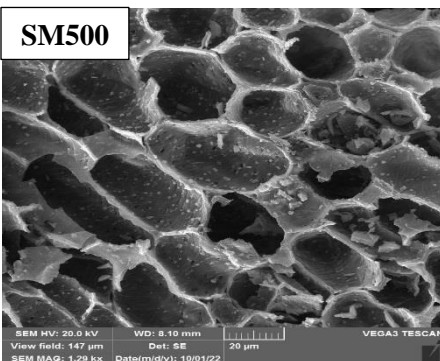
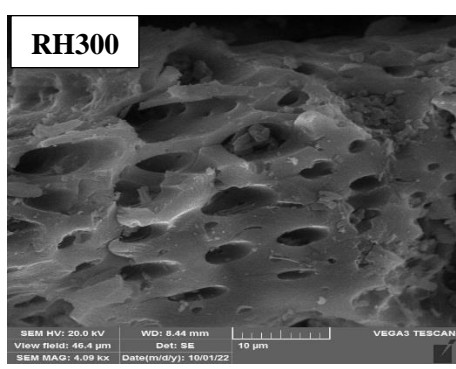
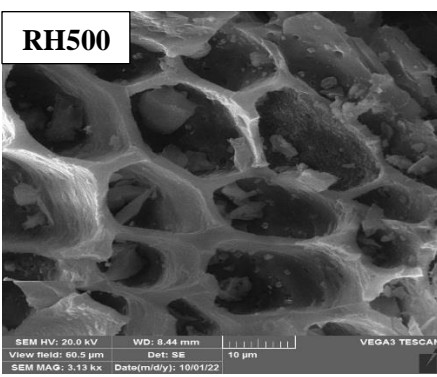

287

**Fig. 3.** SEM images of the biochars. Notes: SM300, sheep manure biochar produced at 300 °C; SM500, sheep manure biochar produced at 500 °C; RH300, rice husk biochar produced at 300 °C; RH500, rice husk biochar produced at 500 °C.

### 3.4 Nickel content in various soil chemical fractions after the application of Si levels and biochars

The interaction of treatments (biochars and Si levels) had a statistically significant effect (P<0.01) on Ni concentration in all the soil chemical fractions, except for the Car fraction. Whereas the main effects of individual treatments (biochars and Si levels) on Ni concentration in all the soil chemical fractions were significant. The Ni concentration in the WsEx fraction was significantly reduced by the application of Si levels ($S_0$ to $S_2$) from 6.07 mg Ni kg$^{-1}$ to 5.17 mg Ni kg$^{-1}$ by 14.8% (Table 4). Among the biochar treatments, the greatest decrease in Ni content in the WsEx fraction compared to the control was due to SM500 from 6.04 mg Ni kg$^{-1}$ to 5.01 mg Ni kg$^{-1}$ by 17%, while the RH300 treatment had no significant effect (Table 4). The interaction effects of treatments indicated that the lowest Ni content in the WsEx fraction was due to the combined treatment of SM500+$S_2$ (4.04 mg Ni kg$^{-1}$ soil) (Table 4). The combined treatment of $S_2$ and SM biochars had strong synergistic effect on reducing Ni content in the WsEx fraction (23-36% reduction) compared to the sum of the treatments alone (13-15% reduction) (Fig. 4). Whereas this synergistic effect of the combined treatments was not evident for the RH biochars (Fig. 4). There was a negative correlation between Ni content in the WsEx fraction and soil pH ($r = -0.66, p < 0.01$) indicating that the reduction in Ni content in the WsEx fraction was strongly linked to the increase in soil pH due to the amendments (Supplementary information). Previous studies have also shown that application of biochars and silicates

result in increases in soil pH, thus reducing the bioavailability of PTEs and their conveyance to plant roots (Shen et al., 2020; Ma et al., 2021). Among the applied biochars, the maximum pH and ash content (Table 3) and calcite (lime) content (Fig. 2) were attributed to the SM500 biochar. Therefore, the combined SM500+$S_2$ was most effective at reducing Ni content in the WsEx fraction, likely due to the higher alkalinity and ash content of SM500 promoting Ni precipitation and adsorption (Sachdeva et al., 2023). The increase in soil pH due to Na metasilicate also likely enhanced the precipitation of Ni in the forms of Ni silicate and hydroxide. Addition of Si in the form of Na metasilicate increases soil pH due to the hydrolysis of the silicate anion in soil solution which generates hydroxyl ions (Ma et al., 2021).

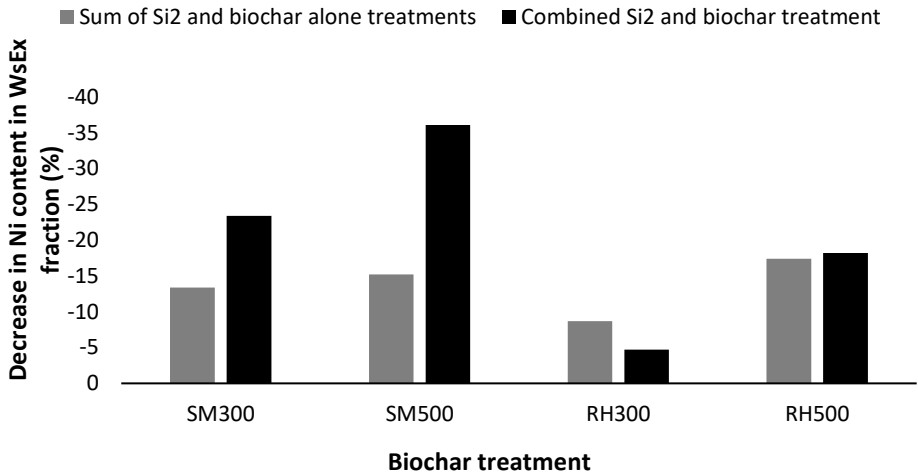

**Fig. 4.** Comparison of the effect of sum of the $Si_2$ and biochar alone treatment versus the combined $Si_2$ and biochar treatments on the reduction of Ni content in the WsEx fraction. Notes: SM300, sheep manure biochar produced at 300 °C; SM500, sheep manure biochar produced at 500 °C; RH300, rice husk biochar produced at 300 °C; RH500, rice husk biochar produced at 500 °C.

The reduced effectiveness of biochars produced at 300 °C, as compared to those produced at 500 °C, in decreasing Ni content in the WsEx fraction could also be attributed to the lower rates of microbial oxidation and mineralization of RH500 and SM500, which is reflected in their greater environmental stability (as indicated by the lower H/C mole ratio values) (Table 3). Consequently, biochar produced at 500 °C may not provide sufficient acidic carboxyl functional groups to the soil to stimulate SOM decomposition, leading to a greater increase in soil pH (Sun et al., 2023). According to Zhu et al. (2015), the addition of wine lees-based biochar (a material from a wine processing factory) to a heavy metal-contaminated soil (at levels of 0.5% and 1% w/w) resulted in an increase in soil pH and a decrease in Ni content in the WsEx fraction.

**Table 4**

Effects of biochars and Si application levels on the Ni concentration (mg kg$^{-1}$) in different soil chemical fractions after corn cultivation.

| | C | SM300 | SM500 | RH300 | RH500 | Mean |
|---|---|---|---|---|---|---|
| | | | WsEx | | | |
| $S_0$ | 6.32 a | 6.02 a-c | 5.91 bc | 6.31 a | 5.77 c | **6.07 A** |
| $S_1$ | 6.03 a-c | 5.37 d | 5.09 de | 6.25 ab | 5.28 d | **5.60 B** |
| $S_2$ | 5.77 c | 4.84 e | 4.04 f | 6.02 a-c | 5.17 de | **5.17 C** |
| Mean | **6.04 A** | **5.41 B** | **5.01 C** | **6.20 A** | **5.41 B** | |
| | | | OM | | | |
| $S_0$ | 9.72 a | 10.6 a | 8.04 d-f | 10.0 a | 9.02 b | **9.40 A** |
| $S_1$ | 9.60 a | 9.75 a | 7.16 g | 8.62 b-d | 8.70 bc | **8.76 B** |
| $S_2$ | 8.11 c-f | 7.94 ef | 7.12 g | 8.30 c-e | 7.63 fg | **7.82 C** |
| Mean | **9.14 A** | **9.28 A** | **7.44 C** | **8.99 A** | **8.44 B** | |
| | | | MnOx | | | |
| $S_0$ | 11.6 a | 3.77 kl | 5.99 f | 4.69gh | 9.71 c | **7.15 A** |
| $S_1$ | 10.3 b | 3.50 l | 5.00 g | 4.57 hi | 8.93 d | **6.48 B** |
| $S_2$ | 10.3 b | 2.98 m | 4.28 ij | 3.96 jk | 7.94 e | **5.89 C** |
| Mean | **10.7 A** | **3.42 E** | **5.09 C** | **4.41 D** | **8.86 B** | |
| | | | AFeOx | | | |
| $S_0$ | 11.1 ef | 10.4 g | 11.8 d | 11.0 fg | 11.7 de | **11.2 C** |
| $S_1$ | 12.2 b-d | 10.7 fg | 12.0 cd | 12.2 b-d | 12.7 bc | **12.0 B** |
| $S_2$ | 12.8 b | 12.2 b-d | 12.2 b-d | 12.3 b-d | 14.2 a | **12.7 A** |
| Mean | **12.1 B** | **11.1 C** | **12.0 B** | **11.8 B** | **12.9 A** | |
| | | | CfeOx | | | |
| $S_0$ | 77.3 f | 78.0 f | 84.0 cd | 84.7 cd | 79.6 ef | **80.7 C** |
| $S_1$ | 77.9 f | 82.2 de | 86.3 bc | 85.1 b-d | 83.6 cd | **83.0 B** |
| $S_2$ | 79.9 ef | 85.5 bc | 87.9 ab | 85.7 bc | 90.4 a | **85.9 A** |
| Mean | **78.4 C** | **81.9 B** | **86.0 A** | **85.2 A** | **84.5 A** | |
| | | | Res | | | |
| $S_0$ | 200 c-e | 207 a | 200 c-e | 196 f | 198 d-f | **200 A** |
| $S_1$ | 200 c-e | 204 b | 200 c-e | 197 ef | 195 f | **199 A** |
| $S_2$ | 200 cd | 204 b | 201 c | 199 c-e | 190 g | **199 A** |
| Mean | **200 B** | **205 A** | **200 B** | **198 B** | **195 BC** | |

Notes: C, control; SM300, sheep manure biochar produced at 300 °C; SM500, sheep manure biochar produced at 500 °C; RH300, rice husk biochar produced at 300 °C; RH500, rice husk biochar produced at 500 °C; $S_0$, without Si addition; $S_1$, application of 250 mg Si kg$^{-1}$ soil; $S_2$, application of 500 mg Si kg$^{-1}$ soil. WsEx, water soluble and exchangeable fraction; OM, organic fraction; MnOx, bound to manganese oxides; AfeOx, bound to amorphous iron oxides; CfeOx, bound to crystalline iron oxides; Res, residual fraction; MF, mobility factor.
*Numbers followed by same letters in each column and rows, in each section, are not significantly (P<0.05) different

As there was no significant interaction effect between biochar type and Si levels on the Ni concentration in the Car fraction, only the significant individual main effects of biochar and Si levels are shown in Fig. 5. Changing the Si application levels from $S_0$ to $S_2$ significantly decreased the Ni content in the Car fraction by 11.7% (from 15.2 mg Ni kg$^{-1}$ soil to 13.5 mg Ni kg$^{-1}$ soil) (Fig. 5). The decrease in the concentration of Ni in the Car form with an increase in the Si levels could potentially be explained by the competition between silicate (SiO$_4^{-4}$) and carbonate ions for binding with Ni$^{2+}$ ions in the soil solution (Sparks et al., 2022). The SM biochars had no significant effect on the Ni concentration in the Car fraction whereas addition of RH biochars led to a significant increase in the Ni concentration in this fraction (Fig. 5). Ippolito et al. (2017) found

that addition of two biochars (pine [*Pinus contorta*] and tamarisk [*Tamarix* spp.]) to a soil contaminated by mining activities caused a significant increase in the soil Cd content bound to carbonates. They concluded that the reduction in Cd bioavailability may have been due to the ability of biochar to raise soil pH levels and induce the precipitation of $CdCO_3$. Similarly, Yuan et al. (2011) proposed that the decrease in bioavailability of PTEs in soil amended by biochars derived from different crop residues might have been caused by the creation of metal-carbonate species and carbonate-surface functional group reactions, which could function as a mechanism for sequestration.

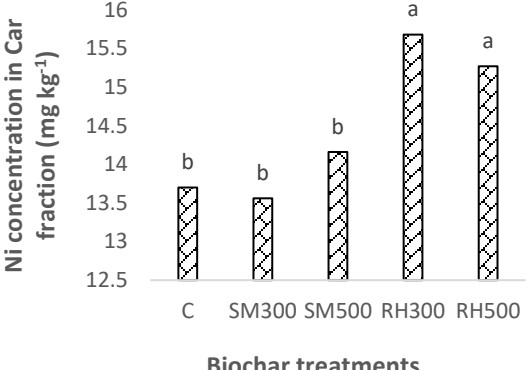 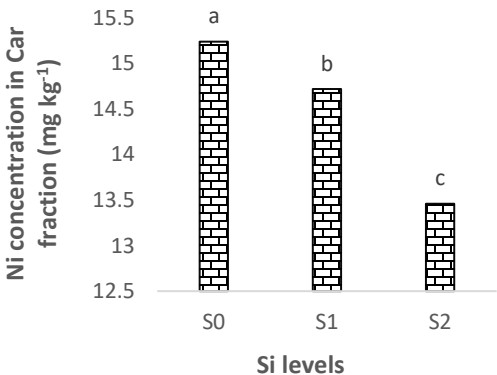

**Fig. 5.** Main effects of (a) biochar type and (b) Si application levels on the Ni concentration (mg kg$^{-1}$) in the Car fraction after corn cultivation. Notes: C, control; SM300, sheep manure biochar produced at 300 °C; SM500, sheep manure biochar produced at 500 °C; RH300, rice husk biochar produced at 300 °C; RH500, rice husk biochar produced at 500 °C; S$_0$, without Si addition; S$_1$, application of 250 mg Si kg$^{-1}$ soil; S$_2$, application of 500 mg Si kg$^{-1}$ soil. * Numbers followed by same letters in each section, are not significantly (P<0.05) different.

The biochars produced at 300 °C had no significant effect on the Ni content in the OM fraction compared to control, while the biochars generated at 500 °C significantly decreased it (Table 4). The greatest reduction in Ni content in the OM fraction (18.6%) was found to be in that which underwent the SM500 treatment. Lu et al. (2017) explored how the application of bamboo and rice straw biochars with varying mesh sizes (0.25 and 1 mm) and at three different levels (0, 1, and 5% w/w) affected the distribution of Cd in a contaminated sandy loam soil, using the BCR (Bureau Communautaire de Référence) sequential extraction method. In contrast to the present study, they reported that the biochars increased the concentration of Cd in the OM fraction, and this was closely related to the increase in Cd immobilization. In another study, the application of sheep manure biochar produced at 500 °C at the rate of 3% wt. to a Cd-contaminated calcareous soil resulted in a significant increase in Cd content in the OM fraction, whereas the addition of other biochar treatments (wheat straw, corn straw, rice husk, licorice root pulp) caused a significant decrease in the concentration of Cd in the OM fraction when compared to the control soil (Boostani et al., 2018a). In the study conducted by Boostani et al. (2018), the reduction in Cd concentration in the OM fraction as affected by application of rice husk biochar is in line with our results,

however; the increase in Cd content in the OM form with addition of sheep manure biochar is in contrast with the result of the present study. These observations indicate that in addition to the characteristics of biochar and the level of its application (Lu et al., 2017), soil characteristics (calcium carbonate percentage, soil texture, etc.) and the type of heavy metal can also have a substantial role in the binding of PTEs to soil organic matter. Changing the Si levels from $S_0$ to $S_2$ reduced the Ni concentration in the OM fraction by 16.8% (Table 4). Ma et al. (2021) reported that the application of Si to cultivated soils significantly reduced soil organic matter content, which could explain why the Ni concentration in OM fraction was reduced in the present study. They indicated that Si facilitates the decomposition of organic matter by enhancing soil pH. In this study, the interaction effects of biochars and Si levels showed that the lowest Ni content in the OM fraction was due to the combined treatment of SM500+$S_2$ (7.12 mg Ni $kg^{-1}$ soil), which was equal to a 26.7% decrease compared to the control (CS$_0$) (9.72 mg Ni $kg^{-1}$ soil) (Table 4).

All the biochar treatments caused a significant decrease in Ni content in the MnOx fraction compared to the control, with the greatest reduction caused by the SM300 treatment at 52.6% (Table 4). The lower temperature biochars were more effective than the higher temperature biochars in decreasing the Ni content in the MnOx fraction (Table 4). In agreement with the present study, Boostani et al. (2023c) observed that biochars produced from cow manure, municipal solid waste and licorice root pulp at a lower pyrolysis temperature (300 °C) decreased the Ni content in the MnOx fraction to a greater extent than those prepared at a higher temperature (600 °C). Hydrophobicity of biochar is decreased with increasing pyrolysis temperature (Kameyama et al., 2019). It is well known that, at the same soil water content, the water content of soil pores treated with biochars produced at lower pyrolysis temperatures is higher due to lower absorption of water by the biochars. Therefore, in soils with high soil pore water content and low oxygen conditions, the concentration of MnOx is decreased due to chemical reduction, while concomitantly, the Mn concentration in the exchangeable and water-soluble are increased (Sparrow and Uren, 2014). Furthermore, increasing the Si application levels from $S_0$ to $S_2$ significantly decreased the Ni content in the MnOx fraction by 17.6% (Table 4). Compared to the control, which had the highest concentration of Ni in the MnOx fraction (11.58 mg Ni $kg^{-1}$ soil), the greatest interactive effect in the reduction of this fraction was related to the combined SM300+$S_2$ (2.98 mg Ni $kg^{-1}$ soil) treatment by 3.8-fold (Table 4). The concentration of Ni bound to the AFeOx and CFeOx fractions was significantly increased by application of Si levels from $S_0$ to $S_2$ by 13.6% and 6.5%, respectively (Table 4). Belton et al. (2012) demonstrated that exogenous silicon application resulted in the attachment of silicate to the surface of iron oxide in the form of a polymer. Following the complexation of ferrosilicon, a significant number of negatively charged functional groups, including silanol, were formed. These groups provided numerous adsorption sites for PTEs, ultimately reducing their bioavailability (Belton et al., 2012). In general, all the biochars caused a significant increase in Ni content in the form of CFeOx, and there were no significant differences among the SM500, RH300 and RH500 treatments (Table 4). However, only the RH500 treatment increased the Ni concentration in the AFeOx fraction compared to control (Table 4). Among all the biochars, only the SM300 resulted in a significant increase in the Ni concentration in the Res fraction compared to the control (Table 4). The main (mean) effects of Si application showed that increasing the Si levels from $S_0$ to $S_2$ had no statistically significant effect on the Ni content in the Res fraction (Table 4).

Mailakeba and Bk (2021) studied the addition of kunai grass biochar (0.75%) to a soil with different Ni contamination levels (0, 56, 100, and 180 mg Ni $kg^{-1}$ soil). They found that the

application of the grass biochar increased Ni content in the Res fraction and reduced Ni in the other fractions in the soil. In another study, Boostani et al. (2023c) demonstrated that the application of biochars (cow manure, municipal compost and licorice root pulp each at 3%(w/w)) to a Ni-contaminated soil increased Ni concentration in the fractions of OM and Res, and decreased Ni content in the fractions of WsEx, Car, and Fe/Mn oxide. Whereas, Boostani et al. (2023b) found that the application of manure and compost biochars (3% w/w) to Pb-contaminated soil did not significantly affect the content of Pb in the Res fraction but did decrease the WsEx fraction. Therefore, it seems that the effect of biochars on the changes of chemical fractions of PTEs in soil depends on the raw materials and production conditions of the biochar, the soil application levels, type of PTEs, the degree of soil contamination with PTEs, the selection of sequential extraction procedure and the soil properties (Mailakeba and Bk, 2021; Boostani et al., 2023b, a; Boostani et al., 2021).

In summary, the application of biochars in the present study resulted in the transfer of Ni in soil from more bioavailable and mobile fractions (WsEx, MnOx, OM) into other stable fractions (AFeOx and CFeOx). These changes were more evident in the WsEx fraction when SM biochars were applied in conjunction with Si (23-36% reduction in Ni content in the WsEx fraction compared to 13-15% when applied alone), indicating that the simultaneous application of these two substances is much more effective than applying them separately.

3.5 Ni concentration in corn (*Zea mays L.*) shoots as affected by treatments

The main effects of biochars, Si application levels and their interactive effects were statistically significant ($P < 0.01$) on the Ni concentration in corn shoots. The main (mean) effects of Si application levels showed that changing the Si levels from $S_0$ to $S_2$ resulted in 32% decrease in the Ni concentration in shoots from 8.56 mg Ni kg$^{-1}$ dry matter (DM) to 5.82 mg Ni kg$^{-1}$ DM (Table 5). In addition, the Ni concentration in shoots was significantly decreased by application of all the biochar treatments compared to the control (with no biochar addition) (Table 5). The interactive effects of treatments indicated that the lowest Ni concentration in shoots was observed in the combined treatment of SM500+$S_2$ (4.45 mg Ni kg$^{-1}$ DM), which showed a 57.2% decrease compared to the control (CS$_0$: without Si and biochar addition) (10.4 mg Ni kg$^{-1}$ DM) (Table 5). The Ni content in shoots had a significant and positive correlation with the Ni content in the WsEx fraction ($r = +0.62, P < 0.01$) while there were a significant and negative correlation between the soil pH ($r = -0.60, P < 0.01$) and Ni content in the CFeOx fraction ($r = -0.50, P < 0.01$) (Supplementary information). This indicates that the application of Si and biochar can reduce the Ni content in shoots by increasing soil pH and, as a result, reducing the amount of Ni in the fraction of WsEx and increasing the Ni content attached to the CFeOx fraction. Boostani et al. (2019a) reported a significant reduction in the concentration of Ni in spinach (*Spinacia oleracea* L.) shoots due to the application of rice husk and licorice root pulp biochars (2.5% w/w) in a Ni-contaminated calcareous soil. Additionally, they reported that the biochars produced at 350 °C were more effective at reducing crop Ni uptake and promoting plant growth than the biochars produced at 550 °C. The most significant factors that contribute to the uptake reduction of PTEs by plants in contaminated soils that have been amended with biochars include adsorption of heavy metals by surface functional groups , increased soil pH, reducing the mobility of PTEs by changing soil redox conditions, improved physical and biological properties of the soil, changes in the activity levels of antioxidant enzymes, and a decrease in the transfer of PTEs to the plant shoots (Zeng et al.,

2018; Rizwan et al., 2016). Several studies have investigated the effect of Si application on Ni concentration in shoots and other heavy metals in various plant species. Khaliq et al. (2016) observed a notable increase Ni concentration and accumulation within the leaf, stem, and roots of cotton after Ni application. Whereas, Si application was observed to induce a significant reduction in Ni concentrations across these respective plant components. In another study, Maryam et al. (2024) concluded that addition of Si caused an increase in the growth indices of maize through reducing the Pb concentration in shoots. One possible explanation for the reduction of Ni concentration in shoots is that Si can compete with Ni for uptake by plant roots. Silicon has a similar ionic radius to Ni, which means that it can occupy the same binding sites on root cell membranes and reduce the uptake of Ni. Additionally, Si can induce the expression of genes that are involved in Ni transport and homeostasis, which may contribute to the reduced Ni concentration in shoots (Hossain et al., 2012; Liang et al., 2005).

**Table 5**
Ni concentration (mg Ni kg$^{-1}$ DM) in corn shoots as affected by biochars and Si application levels.

|  | C | SM300 | SM500 | RH300 | RH500 | Mean |
|---|---|---|---|---|---|---|
| $S_0$ | 10.4 a | 7.35 bc | 9.85 a | 7.55 bc | 7.65 b | **8.56 A** |
| $S_1$ | 7.65 b | 6.90 bc | 6.60 cd | 7.05 bc | 7.35 bc | **7.11 B** |
| $S_2$ | 7.20 bc | 5.05 ef | 4.45 f | 5.80 de | 6.60 cd | **5.82 C** |
| Mean | **8.41 A** | **6.43 C** | **6.96 BC** | **6.80 BC** | **7.20 B** | |

Notes: C, control; SM300, sheep manure biochar generated at 300 °C; SM500, sheep manure biochar generated at 500 °C; RH300, rice husk biochar produced at 300 °C; RH500, rice husk biochar produced at 500 °C; $S_0$, without Si application; $S_1$, addition of 250 mg Si kg$^{-1}$ soil; $S_2$, addition of 500 mg Si kg$^{-1}$ soil. Numbers followed by same letters in each section, are not significantly (P<0.05) different.

3.6 Ni desorption in soil as affected by Si application levels and biochars

The cumulative Ni desorption (extracted by DTPA) in the soil as a function of time are shown in Figure 6. The release of Ni from the soil initially proceeded at a much higher rate during the first hour, and then proceeded at a much slower rate during the next 24 hours, as illustrated by the trend-line in Figure 6. This two-stage process of releasing heavy metals from soil has also been reported by other researchers (Sajadi Tabar and Jalali, 2013; Boostani et al., 2023b). It is likely that the first stage of Ni release is related to forms that are less strongly attached to soil particles, including WsEx and Car, while the second stage of Ni desorption is likely from soil chemical fractions with less bioavailability, such as FeOx and Res (Saffari et al., 2015). In general, the amount of Ni desorption in the soil was reduced by addition of biochars and Si levels (Fig. 6). In addition, the effects of biochars produced at the higher pyrolysis temperature (500 °C) on reducing Ni release was more than those generated at the lower pyrolysis temperature (300 °C) in the soil. The highest amount of Ni release was due to the control (CS$_0$: without biochar and Si application) (37.84 mg Ni kg$^{-1}$ soil) while the lowest was observed in the combined application of SM500 and S$_2$ (31.13 mg Ni kg$^{-1}$ soil) treatment.

498

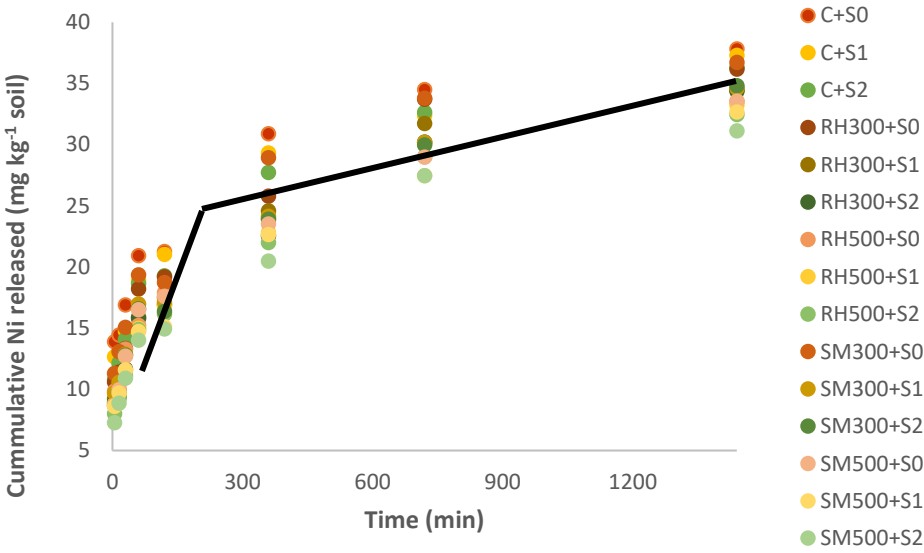

499

**Fig. 6.** Cumulative Ni desorption (extracted by DTPA) (mg kg$^{-1}$) in the soil as affected by different treatments. Notes: C, control; SM300, sheep manure biochar produced at 300°C; SM500, sheep manure biochar produced at 500°C; RH300, rice husk biochar produced at 300°C; RH500, rice husk biochar produced at 500°C; S$_0$, without Si addition; S$_1$, application of 250 mg Si kg$^{-1}$ soil; S$_2$, application of 500 mg Si kg$^{-1}$ soil.

3.7 Fitting of Ni release data to kinetic models

The Ni release data during 24 hours for all the biochar and Si treatments were evaluated by seven different kinetic models (Table 6). The effectiveness of the various kinetic models to describe the observed Ni desorption in the soil was analyzed by considering the coefficient of determination ($R^2$) and standard error of estimate (SEE), so that the highest value of the $R^2$ and the lowest value of the SEE were set as the criteria. As seen in Table 6, the order kinetic models did not adequately describe Ni release in the soil, and with the increase in the order of the kinetic model (from zero to third), the value of the $R^2$ decreased. This has also been found by other researchers for the release of heavy metals from soil (Boostani et al., 2019b; Ghasemi-Fasaei et al., 2006). Whereas, the non-order kinetic models, including power function, parabolic diffusion and simple Elovich, acceptably described the Ni release in the soil amended with various treatments (Table 6). Among them, the power function model was the best according to the highest value of $R^2$ (0.98) and the lowest value of SEE (0.055). Boostani et al. (2018a) also reported that the power function was the best kinetic model to describe Cd desorption in a Cd-contaminated soil treated with biochars and zeolite.





**Table 6**
The range of coefficients of determination ($R^2$) and standard error of estimate (SEE) of applied kinetic models to all the soil treatments.

| Kinetic models | $R^2$ | | SEE | |
|---|---|---|---|---|
| | Range | Mean | Range | Mean |
| Zero order | 0.79-0.87 | 0.80 | 3.36-4.67 | 3.67 |
| First order | 0.69-0.75 | 0.75 | 0.22-0.29 | 0.25 |
| Second order | 0.53-0.61 | 0.52 | $(1.10-2.60) \times 10^{-2}$ | $1.80 \times 10^{-2}$ |
| Third order | 0.39-0.51 | 0.41 | $(1.30-5.20) \times 10^{-3}$ | $3.00 \times 10^{-3}$ |
| Parabolic diffusion | 0.94-0.98 | 0.96 | 1.26-2.44 | 1.85 |
| Power function | 0.97-0.99 | 0.98 | 0.05-0.06 | 0.05 |
| Simple Elovich | 0.92-0.97 | 0.95 | 2.04-2.78 | 2.50 |


3.8 Using the parameters of power function model to investigate the effect of treatments on Ni
desorption in soil

528        As the power function model ($q = at^b$) described the Ni release data the best, its
parameters (a and b) were used to investigate the effect of biochar application and Si levels on the
release of Ni in the Ni-contaminated soil (Table 7). The main effects of biochars and Si levels and
their interactions on the 'a' and 'b' parameters were significant ($P < 0.01$). As Dang et al. (1994)
reported, in this kinetic model, a decrease in parameter 'a' and an increase in parameter 'b' indicates
a decrease in the rate of heavy metal desorption from the soil. The main effects of the treatments
showed that addition of all 4 biochars caused a significant decrease in the 'a' parameter compared
to the control while the 'b' parameter was significantly increased (Table 7). The same trend was
observed for all the Si treatment levels (Table 7). Therefore, it can be concluded that the use of all
the biochars and Si levels caused a decrease in the rate of Ni release in the Ni-contaminated soil.
Generally, there was a greater decrease in Ni desorption in biochar treatments prepared at the
higher temperature (Table 7). The interactive effects indicate that the most effective combined
treatment in reducing the rate of Ni release in the soil was SM500+$S_2$ which had the lowest value
of parameter 'a' (4.52) and the highest value of parameter 'b' (0.26) among the treatments.

542        If it is differentiated from the power function equation ($q = at^b$) with respect to time (t)
($dq/dt = ab\ t^{b-1}$), when $t = 1$ s $= 0$, the ratio of $dq/dt$ becomes 'ab'. This parameter indicates the
amount of heavy metal desorption in the initial time (Dalal, 1985). The 'ab' parameter was affected
by the application of Si levels and biochars, so that this parameter was significantly decreased
compared to the control with addition of all the biochars (12.4%, 24.2%, 15.4% and 21.3% for the
SM300, SM500, RH300 and RH500, respectively) and Si application levels (13% from $S_0$ to $S_2$),
(Table 7). This finding also confirmed the effect of applied treatments in reducing the amount of
Ni release. The greatest reduction was observed in the combined treatment of SM500+$S_2$ by 33.5%
compared to the control (Table 7).




**Table 7**
The coefficients of the power function model as affected by biochars and Si application levels in a Ni-polluted calcareous soil after corn cultivation.

| | C | SM300 | SM500 | RH300 | RH500 | Mean |
|---|---|---|---|---|---|---|
| | | | $a$ (mg Ni kg$^{-1}$ h$^{-1}$)$^b$ | | | |
| $S_0$ | 9.15 a | 7.39 c | 5.56 gh | 6.49 e | 5.95 f | **6.91 A** |
| $S_1$ | 7.92 b | 6.01 f | 5.23 i | 5.66 g | 5.21 i | **6.00 B** |
| $S_2$ | 6.90 d | 5.39 hi | 4.52 k | 5.22 i | 4.84 j | **5.38 C** |
| Mean | **7.99 A** | **6.27 B** | **5.11 E** | **5.80 C** | **5.34 D** | |
| | | | $b$ (mg Ni kg$^{-1}$)$^{-1}$ | | | |
| $S_0$ | 0.20 g | 0.22 e | 0.25 b | 0.24 c | 0.24 c | **0.23 C** |
| $S_1$ | 0.21 f | 0.24 c | 0.25 b | 0.25 b | 0.25 b | **0.24 B** |
| $S_2$ | 0.23 d | 0.25 b | 0.26 a | 0.26 a | 0.26 a | **0.25 A** |
| Mean | **0.21 C** | **0.24 B** | **0.25 A** | **0.25 A** | **0.25 A** | |
| | | | $ab$ | | | |
| $S_0$ | 1.79 a | 1.68 c | 1.37 h | 1.54 e | 1.41 g | **1.55 A** |
| $S_1$ | 1.69 b | 1.43 f | 1.29 k | 1.41 fg | 1.32 j | **1.42 B** |
| $S_2$ | 1.59 d | 1.37 h | 1.19 m | 1.34 i | 1.27 l | **1.35 C** |
| Mean | **1.69 A** | **1.48 B** | **1.28 E** | **1.43 C** | **1.33 D** | |

Notes: C, control; SM300, sheep manure biochar produced at 300 °C; SM500, sheep manure biochar produced at 500 °C; RH300, rice husk biochar produced at 300 °C; RH500, rice husk biochar produced at 500 °C; $S_0$, without Si addition; $S_1$, application of 250 mg Si kg$^{-1}$ soil; $S_2$, application of 500 mg Si kg$^{-1}$ soil.

* Numbers followed by same letters in each column and rows, in each section, are not significantly (P<0.05) different

The correlation between the parameters of the fitted power function model with Ni content in various soil chemical fractions, Ni concentration in corn shoots and soil pH are shown in Table 8. The $´a´$ and $´ab´$ parameters had a positive correlation with the Ni content in the WsEx, OM and MnOx fractions, while theses parameters had a negative correlation with the Ni content in the AFeOx and CFeOx fractions. This trend was inverse for the $´b´$ parameter of the power function model. These correlations verified that the application of Si and biochar to the Ni-contaminated calcareous soil led to a decrease in the rate and amount of Ni release by reducing the Ni concentration in chemical forms with higher bioavailability including WsEx, OM and MnOx. Furthermore, the $´a´$ and $´ab´$ parameters were negatively correlated with soil pH. Whereas there were positive correlations between these parameters and Ni concentration in shoots (Table 8). These findings once again confirmed that the increase in soil pH due to the application of Si and biochar can cause a decrease in bioavailability of Ni in the soil and, as a result, a decrease in the concentration of Ni in aerial parts of the plant.

**Table 8**
The Pearson correlation coefficients (r) among the power function model parameters (a, b, ab) with Ni content in soil chemical fractions, Ni concentration in corn shoots and soil pH.

| | WsEx | Car | OM | MnOx | AFeOx | CFeOx | Res | Ni content in shoots | Soil pH |
|---|---|---|---|---|---|---|---|---|---|
| a | 0.63** | 0.02ns | 0.70** | 0.53** | -0.44** | -0.80** | 0.27ns | 0.62** | -0.52** |
| b | -0.59** | 0.03ns | -0.68** | -0.54** | 0.46** | 0.83** | -0.28ns | -0.63** | 0.51** |
| ab | 0.68** | 0.04ns | 0.74** | 0.46** | -0.46** | -0.80** | 0.29ns | 0.06** | -0.51** |

Notes: WsEx, water soluble and exchangeable fraction; OM, organic fraction; MnOx, bound to manganese oxides; AFeOx, bound to amorphous iron oxides; CFeOx, bound to crystalline iron oxides; Res, residual fraction.
** and ns indicate significance at the 0.01 probability level and non-significant, respectively.

## 4 Conclusions

The application of biochars and Si in the present study resulted in the transfer of Ni in soil from more bioavailable and mobile fractions (WsEx, MnOx, OM) to more stable forms (AFeOx and CFeOx). These changes were particularly evident in the WsEx fraction when SM biochars were applied in conjunction with silicon, indicating a strong synergistic effect related to soil pH increase. Application of all biochars and Si reduced DPTA-extractable Ni release in the soil, which was most strongly associated with the increase in Ni content in the CFeOx fraction. Application of all the biochars and Si decreased corn Ni uptake, with the combined SM500+$S_2$ being the most effective. The decrease in corn uptake was correlated with the decrease in Ni content in the WsEx fraction and increase in the CFeOx fraction. SM500 was likely the most effective biochar due to its higher alkalinity and ash content, and lower acidic functional group content which enhances Ni sorption reactions with Si. Future research is needed to better understand the mechanisms underlying the interaction effects of Si and biochar application on the distribution of Ni in different soil chemical fractions and to optimize Si application strategies for sustainable Ni management in agricultural and natural ecosystems. It is suggested that the interactive effects of Si and biochar on the Ni content in soil chemical fractions and its release in aged Ni-contaminated soils should also be investigated and compared, as this study was limited to a recently Ni-contaminated soil.

**Authors' Contributions** H.R.B. Conceptualization, Formal analysis, Methodology, Investigation, Validation A.G.H. Writing - Review & Editing M.N. Project administration, Visualization E.B. Review & Editing E.F. Laboratory analyses.

**Financial support.** No funding was received for conducting this study.

**Competing interests.** The contact author has declared that neither they nor their co-authors have any competing interests.

**Data availability.** The data generated in this study are available from the corresponding authors upon reasonable request.

**Disclaimer.** Publisher's note: Copernicus Publications remains neutral with regard to jurisdictional claims in published maps and institutional affiliations.

**Acknowledgements:** This work was supported by College of Agriculture and Natural Resources of Darab, Shiraz University, Darab, Iran.

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
