# Peer review of "Investigating the synergistic potential Si and biochar to immobilize Ni in a Ni-contaminated calcareous soil after Zea mays L. cultivation"

_EGUsphere, 2023_

## Referee Comment (RC2)

**Investigating the synergistic potential Si and biochar to immobilize soil Ni in a contaminated calcareous soil after Zea mays L. cultivation**

**Summary**

Interesting study that could provide useful information in regard to the immobilisation of Ni in agricultural soils. However the manuscript needs a significant re-write before it is in any way acceptable for publication.

**General comments**

- The standard of English is generally good but improvements should be made throughout to ensure smooth reading, especially related to the use of the abbreviation PTEs.
- There is a distinct lack of detail in all sections, all of which would benefit from considerable expansion. The reader should be able to replicate this experiment exactly in the same way the authors carried it out – impossible currently with the paucity of information.
- More thought should be given to presentation and interpretation of results and results should be better compared to available literature. Please don't split tab les over 2 pages, very difficult to review.

**Specific comments**

- More background information is needed throughout the manuscript to enhance understanding of the need for the experiment and also on contamination and soil processes and how these relate to the importance and findings of this study.
- The Introduction needs to be greatly expanded to include more information on why this study is so important, especially for a country like Iran. For example what area of the country is available for agriculture, how much is needed/is it enough? How much is land is contaminated and with what…? Why is Ni particularly relevant?
- Also in the introduction much more detail on the various soil processes/mechanisms/chemistry related to the bioavailability of PTEs, their distribution between various soil fractions and how mechanistically additions/amendments increase or decrease both the bioavailability of PTEs like Ni in soil and the enhancement or otherwise of crop growth.
- Lots of things are stated without giving any examples. E.g. L73 "The quantity and rate of release of soil PTEs from soil particles over time can influence their bioavailability". In what way? Is it the same for all PTEs? How does Ni compare to other PTEs? Are the processes generally the same? How do they differ? Also add details on climate, topography and the variety of soil types in Iran, are the soils studied representative? Coordinates of sampled soils contaminated and otherwise. etc. Maps would be useful – show location of sites in Iran at a minimum.

**Technical corrections**

L 18 Please delete "remains" as other studies have been carried out on this topic. Or be more specific.

L18 Please change to "affects the immobilization of PTEs…

Hereafter read "please" for all changes

L22 Sort out S1 closing bracket

L23 Change included to including

L34 Add "of" between potential and Si

L37 Add "through" between the and consumption

L45 Expand on contaminated sites and what the relevance is – perhaps percentage area of the country, why it is important to use this land. What are the expensive techniques used…

L51 Do you mean "soil stress caused by PTEs?

I will stop with comments on English here – manuscript needs the English to be improved throughout, specifically I would read out the abbreviation for PTEs when rewriting – see previous comment and rearrangement of word order, e.g. L67 should read "in stabilizing PTEs in soils…". L68 reads "Soil potentially toxic elements….", no, "potentially toxic elements in soils…."

L53 Give example of other materials that are not as good for remediation

L55 Silicon itself does not directly increase soil pH. However, silicon can indirectly influence soil pH through various mechanisms – please provide some background / further expansion on this

L71 delete "as"

L73 what characteristics?

L74 – expand on this process – give some examples on how bioavailability can be influenced

L88 – remains – see previous comment L18, or be more specific

L96 polluting? Pollution? What characteristics?

L97 Section 2.1 – add more details on general soil characteristics (how do these compare for example to other Iranian soils and to European agricultural soils) laboratory methods / equipment.

- Also more detail on the method of Boostani et al., readers don't want to have to go searching in other papers to find out what you have done – provide brief description at a minimum. Ni(Cl2) solution, more details on concentration, purity, supplier….
- How were soils sampled? Where, with what? How was representativeness ensured?

L104 Section 2.2 – ditto above, much more detail required. Where did the feedstocks come from? What are they? What weights were used? What kind of mesh? Material? Supplier?

L113 and throughout manuscript - space between all temperatures and °C

L116 contaminated soil samples – this just appears from nowhere. What contaminated soil? From where? Contaminated with what and by how much? How was the contaminated soil

sampled? How many sampling points? Were all the samples combined and then sub-sampled?

Section 2.3 – much more detail required – what are the pots? How many pots? What weight of soil in the pots? Controls? Replicates?.... What required reactions? Why was distilled water used to keep soil at field capacity? Is field capacity representative of Iranian soils? If not how are results extrapolated to reflect average water content in soils in this part of Iran?

Section 2.4 – ditto above sections – provide enough detail that other researchers can repeat exactly what you have done. Details on chemicals, suppliers…

Section 2.5 Ditto above and improve English

Section 2.6 Ditto above and improve English

L153 Discussion not discussions

L155 and on, Section 3, how was SOM quantified? Ditto CEC? This is not mentioned in section 2. Ditto all other determined values listed in Table 2.

L155 What uncontaminated soil?

L165 How exactly was the pH, EC and other parameters listed in Table 3 determined?

L171 Surface functional groups and CEC – add some information on this to the Introduction

L181 Expand on how the ratios indicate what you say here

L182 The results "displayed in" Table 3. Table 3 did not by itself produce results. Where are the uncertainties in Table 3? Are differences in this table significant or not?

L184 indicate not indicated

L185 Insignificant how? Compared to what? As it appears none was detected

L186 – limited yes but more than the RH biochar and is the difference between SM300 and SM500 significant?

L188, Section 3.3 FTIR and SEM come out of nowhere. Details in the methods section please. In this section replace evident with evidenced or rather "as indicated"

L205 Figure 2

L252 Figure 3

Figure 3 – why sum Si and biochar? Why not show them separately?

L 213 Change to "Soil Ni chemical fractions after addition of silicon and biochar"

L215 – significant how? What happened to the fractions? Confusing, please re-write. Actually man paragraphs in this section need to be revised as there are a lot of random observations: Table 4 results are discussed, then Table 3, then Figure 1 without a logical coherent story/interpretation. Again, a better review of the literature in the introduction would help here, i.e. what previous studies have shown when it comes to amendments to soils, and their affect on PTEs and crop production.

L239 Was microbial activity measured? Can you really say with certainty from lower H/C mole ratios that there were lower rates of microbial oxidation?

Figure 4 – Ni not Cd on y axis

Table 4 What are the values in bold in the right hand column?

L252 State from what concentration to what concentration that Ni decreased by (11.7%). Is is Ni-Car or Car-Ni? Figure 4 Cd-Car

L275 rates?

L276 Define BCR

L278 Application rate? A rate implies speed of application

L282 Why the difference between SM biochar and the others? Provide some more detail, mechanisms

L291 ---

L292 rates? The addition of $S_0$ is not an addition – nothing was added – please rephrase

L294 C+$S_0$ is not a treatment -it is a control with no addition – please rephrase

Table 7 - mg Cd kg-1 h-1??

Without going further this manuscript need a significant re-write before it is in any way acceptable for publication. If the authors revise the manuscript taking into account the above comments and apply the same thoroughness to the remainder of the manuscript I will happily review this again from the beginning.

---

## Author Response (AR1)

**RESPONSE TO REVIEWER COMMENTS ON MANUSCRIPT:** Investigating the synergistic potential Si and biochar to immobilize soil Ni in a contaminated calcareous soil after *Zea mays* L. cultivation Egusphere-2023-2687

The authors would like to thank anonymous reviewers for their time, invaluable comments and suggestions for substantially improving this manuscript. Please find detailed responses to each comment below.

**ALL CHANGES ARE INDICATED IN GREEN HIGHLIGHT IN THE REVISED MANUSCRIPT**

**ANONYMOUS REFEREE #1**

1. **L. 61: Please, change "by product" by "byproduct" or "by-product"**

Authors' response: We have made the correction as suggested by the reviewer.

2. **L. 100: "… various physicochemical soil properties". Please, describe which physicochemical soil properties were determined, and the methodology (with references).**

Authors' response: We have added a concise description of each of the methods used to determine the soil properties with references, as also suggested by Reviewer 2.

3. **L. 102: Please, add a brief description of the Boostani et al. (2023c) method. Was the soil incubated with Ni(Cl$_2$)? How long, under what conditions (temperature, darkness, moisture)?**

Authors' response: We have added a concise description of the methodology as suggested by the reviewer, as also suggested by Reviewer 2.

4. **L. 108: Please, briefly describe which chemical characteristics were assessed in biochar, and the methodology (with references).**

Authors' response: We have added a concise description of each of the methods used to determine the biochar properties with references, as also suggested by Reviewer 2.

5. **L. 137: Please, recalculate and replace "r.p.m." by "x g"**

Authors' response: We have recalculated and corrected it as suggested by the respectable reviewer.

6. **Table 2: The Ni total content in soil prior to experiments is missing, i.e., the original Ni total content of the soil.**

Authors' response: We have added the soil total Ni in Table 2 as suggested by the respectable reviewer.

7. **L. 188: FTIR and SEM analysis should be in Material and Methods**

Authors' response: We have added a concise description of the biochar properties that were determined including FTIR and SEM to the Materials and Methods section as suggested by the reviewer.

8. **L. 192: It's Table 3, isn't it?**

Authors' response: We have corrected the Table number as pointed out by the reviewer.

9. **L. 205: It's Figure 2, isn't it?**

Authors' response: We have corrected the Figure number as also pointed out by Reviewer 2.

**10.L. 231: Why was the soil pH not measured after the experiments? If possible, determine and add to the manuscript these data to support your hypothesis.**

Authors' response: Soil pH was measured after the experiments, as the correlation between soil pH and was stated in Line 226 of the original manuscript. We have added the soil pH data to the manuscript **supplementary information** as requested by the reviewer.

**11.Same comment for L. 283 – 284 ("It has been shown that the application of Si to cultivated soils resulted in a reduction of soil organic matter content".**

Authors' response: Unfortunately, it is currently not possible to measure the amount of soil organic matter.

**12.L. 232: Sachdeva et al., 2023: Full citation missing in the reference list.**

Authors' response: This reference was added to the reference list.

**13.Figure 4: Please, revise the Y-axis "Cd-Car"**

Authors' response: The Y-axis of Fig. 4 has been corrected to Car-Ni as also pointed out by Reviewer 2.

**14.Why was the Ni-Car fraction represented in a separate Figure and no with the other values in Table 4?**

Authors' response: This is because, unlike the other soil Ni fractions, there was not a significant statistical interaction effect between biochar treatments and Si levels on the Ni-Car fraction. Therefore, the main effects of biochar and Si levels were shown separately in Figure 4 a and b, respectively.

**15.L. 262: If possible, add a reference which justified the hypothesis "the decrease in the concentration of Ni in the carbonate … Ni+2 ions in the soil solution"**

Authors' response: We have added a reference as suggested by the respectable reviewer.

**16.L. 273 – 278: "Lu et al. (2017) explored … rate of the biochar". What do you mean by this study? It is not very clear in the text, as they compare two types of biochar, but both of plant origin, whereas in the manuscript the comparison is between manure biochar and rice husk biochar. Furthermore, no specific (and related) conclusions of the study are given, only general information.**

Authors' response: Thank you for pointing out this error. We have added the relationship of the previous studies results with the results of our study, and we have added a conclusion for the study.

**17.L. 278 – 282: "In another study, the application … control soil (Boostani et al., 2018)". The results obtained in this study are very interesting to discuss your results. However, the results obtained in this manuscript were not specifically compared and discussed with this study.**

Authors' response: We have now specifically compared and discussed the results of the Boostani et al. (2018) with our results as suggested by the respectable reviewer.

**18.L. 289 – 290: Why was the MnOx-Ni fraction more affected by lower T biochars than higher T biochars?**

Authors' response: A discussion of similar results to our study was added to the manuscript. Also, the possible mechanism in relation to it was explained. Please see the revised manuscript.

**19.L. 326; Please, revise the italics (*Zea mays* L.)**

Authors' response: We have corrected the italics as pointed out by the reviewer.

**20.Table 5: Please, add the meaning of the letters accompanying Ni concentrations**
Authors' response: The meaning of the letters has been added to the caption of Table 5, as suggested by the reviewer.

**21.331: RH300 also statistically differs (Table 5)**
Authors' response: Application of all the biochars caused a significant decrease in the Ni-shoot content compared to the C (Table 5). The sentence ˝**Only the RH500 and SM300 treatments differed statistically from each other**˝ was removed to avoid ambiguity.

**22.Why SM300 statistically differs from C, but SM500 does not? (Table 5)**
Authors' response: The significant letter associated to the control treatment (A) was different with the significant letters attributed to the SM500 (BC) and SM300 (C). Thus, both of them (SMB300 and SM500) had statistically different compared to control. Please see Table 5.

**23.348 – 349: "Several studies have investigated the effect of soil Si application on shoot Ni concentration in various plant species" Which studies? Please, add them on the manuscript.**
Authors' response: We added two studies about the effects of Si on reducing the shoot Ni and Pb concentration in two different plants. Please see the revised manuscript.

**24.387: Heavy elements? Do you mean heavy metals?**
Authors' response: Thank you for pointing out this error, it has been corrected to heavy metal.

**25.574: Please, revise the italics (*Brassica…*), Also in L. 514, 505, 558, 514, etc. Please, revise the italics along the manuscript (particularly in the reference list)**
Authors' response: We have corrected all instances of where scientific plant names were not italicized as pointed out by the reviewer.

**ANONYMOUS REFEREE #2**

**Summary**
Interesting study that could provide useful information in regard to the immobilisation of Ni in agricultural soils. However, the manuscript needs a significant re-write before it is in any way acceptable for publication.

**General comments**
   1. **The standard of English is generally good but improvements should be made throughout to ensure smooth reading, especially related to the use of the abbreviation PTEs.**
Authors' response: The manuscript has been thoroughly revised according to the reviewers' comments, and English improved.

   2. **There is a distinct lack of detail in all sections, all of which would benefit from considerable expansion. The reader should be able to replicate this experiment exactly in the same way the authors carried it out – impossible currently with the paucity of information.**

Authors' response: The manuscript materials and methods section has been thoroughly revised according to the reviewers' comments to improve description of methodology used. See responses to specific comments of Reviewer 1 and 2.

3. **More thought should be given to presentation and interpretation of results and results should be better compared to available literature. Please don't split tables over 2 pages, very difficult to review.**

Authors' response: The manuscript Results and discussion section has been thoroughly revised according to the reviewers' comments to provide more specific explanations and discussions of the results in comparison to the scientific literature. See responses to specific comments of Reviewer 1 and 2. Care has been taken to ensure that tables are not split over 2 pages.

**Specific comments**

4. **More background information is needed throughout the manuscript to enhance understanding of the need for the experiment and also on contamination and soil processes and how these relate to the importance and findings of this study. The Introduction needs to be greatly expanded to include more information on why this study is so important, especially for a country like Iran. For example what area of the country is available for agriculture, how much is needed/is it enough? How much is land is contaminated and with what…? Why is Ni particularly relevant?**

Authors' response: Thank you for your suggestions. We added latest information available about of Ni content in agricultural soils of Iran to the introduction section. Please see the revised manuscript.

5. **Also, in the introduction much more detail on the various soil processes/mechanisms/chemistry related to the bioavailability of PTEs, their distribution between various soil fractions and how mechanistically additions/amendments increase or decrease both the bioavailability of PTEs like Ni in soil and the enhancement or otherwise of crop growth.**

Authors' response: We added more specific mechanistic information on how biochar reduces Ni bioavailability in soils to the Introduction section. Please see the revised manuscript.

6. **Lots of things are stated without giving any examples. E.g. L73 "The quantity and rate of release of soil PTEs from soil particles over time can influence their bioavailability". In what way? Is it the same for all PTEs? How does Ni compare to other PTEs? Are the processes generally the same? How do they differ? Also add details on climate, topography and the variety of soil types in Iran, are the soils studied representative? Coordinates of sampled soils contaminated and otherwise. Etc. Maps would be useful – show location of sites in Iran at a minimum.**

Authors' response:  Regarding PTEs release from soil, for clarity and better understanding, some changes were made in the Introduction section as suggested by the respectable reviewer. Also, a brief description of the dominant soils of Iran with their characteristics was added in Section 3.1. (Soil characteristics). As mentioned in Materials and Methods section 2.1; we collected an uncontaminated composite soil sample from research farm of the College of Agriculture and Natural Resources in Darab, southern Iran (28° 45′ 0.99″ N 54° 26′ 52.14″ E, Elevation 1105 m) and then, they were artificially contaminated by nickel. At the same time, we added the geographical coordinates of the sampling area to the manuscript. It should be noted that a comprehensive map of places contaminated with toxic elements has not yet been prepared in Iran.

**Technical corrections**

7. **L 18 Please delete "remains" as other studies have been carried out on this topic. Or be more specific.**

Authors' response: We have rewritten the statement to be more specific and highlight the novelty of this study. It now reads: "No previous studies have examined the potentially synergistic effect of Si and biochar on soil Ni chemical fractions and immobilization."

8. **L18 Please change to "affects the immobilization of PTEs…**

Authors' response: The sentence has been completely rewritten in response to the reviewer's previous comment no. 8.

**Hereafter read "please" for all changes**

9. **L22 Sort out S1 closing bracket**

Authors' response: The correction has been made as suggested by the reviewer.

10. **L23 Change included to including**

Authors' response: The correction has been made as suggested by the reviewer.

11. **L34 Add "of" between potential and Si**

Authors' response: The correction has been made as suggested by the reviewer.

12. **L37 Add "through" between the and consumption**

Authors' response: The correction has been made as suggested by the reviewer.

13. **L45 Expand on contaminated sites and what the relevance is – perhaps percentage area of the country, why it is important to use this land. What are the expensive techniques used…**

Authors' response: In this section, the latest official information reported about nickel content in Iranian soils and the percentage of nickel-contaminated soils was added. Also, the cost methods for remediating contaminated soils were briefly mentioned.

14. **L51 Do you mean "soil stress caused by PTEs?**

Authors' response: We have rewritten the sentence to improve clarity. It now reads: "Applying Si to the soil can enhance plant resistance against biological and non-biological tensions, including physiological stress caused by PTEs in soil"

15. **I will stop with comments on English here – manuscript needs the English to be improved throughout, specifically I would read out the abbreviation for PTEs when rewriting – see previous comment and rearrangement of word order, e.g. L67 should read "in stabilizing PTEs in soils…". L68 reads "Soil potentially toxic elements….", no, "potentially toxic elements in soils…."**

Authors' response: We have corrected all instance of soil PTEs to PTEs in soils as suggested by the reviewer.

16. **L53 Give example of other materials that are not as good for remediation**

Authors' response: the better sentence was replaced to clarify. Please see the revised manuscript.

**17. L55 Silicon itself does not directly increase soil pH. However, silicon can indirectly influence soil pH through various mechanisms – please provide some background / further expansion on this**

Authors' response: We have added at sentence to explain the soil pH increase due to Si application.

**18. L71 delete "as"**

Authors' response: The correction has been made as suggested by the reviewer.

**19. L73 what characteristics?**

Authors' response: some important soil characteristics were added. Please see the revised manuscript.

**20. L74 – expand on this process – give some examples on how bioavailability can be influenced**

Authors' response: Pease see the author response for question 6.

**21. L88 – remains – see previous comment L18, or be more specific**

Authors' response: The sentences have been rewritten as follows to be more specific: "As both biochars and Si are economical and effective soil amendments to reduce plant PTE uptake and stress in contaminated soils, their potential synergistic effect on the immobilization of PTEs in soils should be further investigated. Currently, no previous studies have examined the combined application effects of Si and biochars the chemical fractions and release kinetics of Ni in calcareous soils.

**22. L96 polluting? Pollution? What characteristics?**

Authors' response: The section 2.1 heading has been corrected as follows: "2.1 Soil sampling, characterization and nickel treatment".

**23. L97 Section 2.1 – add more details on general soil characteristics (how do these compare for example to other Iranian soils and to European agricultural soils) laboratory methods /equipment.**

Authors' response: We have added a concise description of each of the standard international methods used to determine the soil properties with references, as also suggested by Reviewer 1.

**24. Also more detail on the method of Boostani et al., readers don't want to have to go searching in other papers to find out what you have done – provide brief description at a minimum. Ni(Cl2) solution, more details on concentration, purity, supplier….**

Authors' response: We have added a concise description of the Ni treatment methodology, as also suggested by Reviewer 1. Also, the characteristics of applied Ni(Cl)$_2$ was added.

**25. How were soils sampled? Where, with what? How was representativeness ensured? L104 Section 2.2 – ditto above, much more detail required.**

Authors' response: Section 2.2 was revised as suggested by the respectable reviewer.

**26. Where did the feedstocks come from? What are they? What weights were used? What kind of mesh? Material? Supplier?**

Authors' response:  The requested information about feedstocks were added to the manuscript in the section 2.2. please see the revised manuscript.

**27. L113 and throughout manuscript – space between all temperatures and °C**

Authors' response: The correction has been made throughout the manuscript as pointed out by the reviewer.

**28. L116 contaminated soil samples – this just appears from nowhere. What contaminated soil? From where? Contaminated with what and by how much? How was the contaminated soil sampled? How many sampling points? Were all the samples combined and then subsampled?**

Authors' response: The term "contaminated" has been replaced by "Ni-treated" to correspond with Section 2.1, where the Ni treatment (contamination) of the soil is explained.

**29. Section 2.3 – much more detail required – what are the pots? How many pots? What weight of soil in the pots? Controls? Replicates?.... What required reactions? Why was distilled water used to keep soil at field capacity? Is field capacity representative of Iranian soils? If not how are results extrapolated to reflect average water content in soils in this part of Iran?**

Authors' response: As suggested by the respectable reviewer, the type of pots, the number of pots and the amount of soil in them were added to the manuscript. Because a rich source of carbon has entered the soil, we gave it two weeks to minimize the activity of the soil microorganisms to prevent competition between the microorganisms and the plants for nitrogen absorption in the early stages of plant growth. We used distilled water to irrigate the pots during the growing season, so that no substances are added to the soil through the irrigation water and the conditions are controlled. At field capacity level, the plant uses the least amount of energy to absorb water, and if we considered a lower soil moisture content, the plant might experience moisture stress in addition to the Ni stress.

**30. Section 2.4 – ditto above sections – provide enough detail that other researchers can repeat exactly what you have done. Details on chemicals, suppliers…**

Authors' response: We added more detailed on the chemical used in each of the sections as suggested by the respectable reviewer. The extraction method of Singh et al. (1988) is a well-known method for soil scientists to separate the forms of heavy elements in soil, and usually it is enough to mention this method in published articles, although Table 1 In short, shows the working method and the chemicals used in it.

**31. Section 2.5 Ditto above and improve English**

Authors' response: We have checked the English grammar again and corrected it. we added more detailed on the chemical used in each of the sections as suggested by the respectable reviewer.

**32. Section 2.6 Ditto above and improve English**

Authors' response: We have checked the English grammar again and corrected it. We added more detailed on the chemical used in each of the sections as suggested by the respectable reviewer.

**33. L153 Discussion not discussions**

Authors' response: The correction has been made as suggested by the reviewer.

**34. L155 and on, Section 3, how was SOM quantified? Ditto CEC? This is not mentioned in section 2. Ditto all other determined values listed in Table 2.**

Authors' response: We have added a concise description of each of the methods used to determine the soil properties with references, as also suggested by Reviewer 1.

**35. L155 What uncontaminated soil?**

Authors' response: The sentence has been rewritten as follows to clarify the intended meaning: "The soil used in the study prior to experimental treatment, exhibited a sandy loam texture and possessed alkaline properties with significant calcium carbonate content, while not being classified as saline (Table 2)."

**36. L165 How exactly was the pH, EC and other parameters listed in Table 3 determined?**

Authors' response: We have added a concise description of each of the methods used to determine the biochar properties with references, as also suggested by Reviewer 1.

**37. L171 Surface functional groups and CEC – add some information on this to the Introduction**

Authors' response: A sentence has been added to the Introduction regarding biochar surface functional groups and CEC.

**38. L181 Expand on how the ratios indicate what you say here**

Authors' response: The sentence has been amended to explain the ratios as follows: "The ratios of H:C and O:C are significant indicators of the aromaticity and polarity of biochars; the lower the ratios the more condensed aromatic C the biochar contains (Chatterjee et al., 2020).

**39. L182 The results "displayed in" Table 3. Table 3 did not by itself produce results.**

Authors' response: We have corrected "of" to "shown in".

**40. Where are the uncertainties in Table 3? Are differences in this table significant or not?**

Authors' response: Usually, the values of the properties related to the amendments are expressed quantitatively and they are not compared statistically. Only in some cases, the standard deviation of each value is given.

**41. L184 indicate not indicated**

Authors' response: The correction has been made as suggested by the reviewer.

**42. L185 Insignificant how? Compared to what? As it appears none was detected**

Authors' response: The sentence has been rewritten as follows to improve clarity: The Ni content in the biochars derived from rice husk was below detection, whereas a limited quantity of Ni was detected in the biochars produced from sheep manure (Table 3).

**43. L186 – limited yes but more than the RH biochar and is the difference between SM300 and SM500 significant?**

Authors' response: Here, only more or less is considered and statistical comparison is not customary for it. Please see the author response for comment 42.

**44. L188, Section 3.3 FTIR and SEM come out of nowhere. Details in the methods section please.**

Authors' response: A description of the FTIR and SEM methodology used to characterize the biochars was added to Section 2.2 as also requested by Reviewer 1.

**45. In this section replace evident with evidenced or rather "as indicated"**

Authors' response: "evident" was replaced by "as indicated" as suggested by the reviewer.

**46. L205 Figure 2**

Authors' response: We have corrected the Figure number as also pointed out by Reviewer 1.

**47. L252 Figure 3**

Authors' response: This particular discussion regarding Ni-Car is correctly referring to Fig. 4, and so we have not changed it.

**48. Figure 3 – why sum Si and biochar? Why not show them separately?**

Author's response: Figure 3 illustrates that the synergistic effect of the combined treatment is substantially greater than simply the sum of the two treatments separately.

**49. L 213 Change to "Soil Ni chemical fractions after addition of silicon and biochar"**

Author's response: The section 3.4 heading has been amended as suggested by the reviewer.

**50. L215 – significant how? What happened to the fractions? Confusing, please re-write.**

Author's response: The sentence has been rewritten to improve clarity. It now reads: "The main effects of treatments (biochars and Si levels) and their interactions had a statistically significant effect ($P<0.01$) on al the soil Ni chemical fractions, except for the Ni-Car fraction, where only the main effects were significant".

**51. Actually many paragraphs in this section (3.4) need to be revised as there are a lot of random observations: Table 4 results are discussed, then Table 3, then Figure 1 without a logical coherent story/interpretation. Again, a better review of the literature in the introduction would help here, i.e. what previous studies have shown when it comes to amendments to soils, and their affect on PTEs and crop production.**

Author's response: Section 3.4 discussions have been revised according to the reviewer's suggestions. Also, we tried as much as possible to add content that would contribute to the richness of the manuscript as long as the introduction section is not too long. Please see the introduction section in revised manuscript.

**52. L239 Was microbial activity measured? Can you really say with certainty from lower H/C mole ratios that there were lower rates of microbial oxidation?**

Author's response: In this experiment, we did not measure microbial activity, and based on the results of Sun et al. (2023) as given in the text of the manuscript, we expressed this mechanism as a possibility.

**53. Figure 4 – Ni not Cd on y axis**

Authors' response: The y-axis has been corrected as also pointed out by Reviewer 1.

**54. Table 4 What are the values in bold in the right-hand column?**

Authors' response: We have added the word "Mean" to the heading of the right-hand column in Table 4.

**55. L252 State from what concentration to what concentration that Ni decreased by (11.7%).**

Authors' response: the Ni concentrations in the CAR form associated with the $S_0$ and $S_2$ treatments were indicated as suggested by the respectable reviewer.

**56. Is Ni-Car or Car-Ni? Figure 4 Cd-Car**

Authors' response: The axis has been corrected to Car-Ni.

**57. L275 rates?**

Authors' response: In agronomy and soil science, an application rate refers to the mass of amendment applied per mass or area of soil.

**58. L276 Define BCR**

Authors' response: BCR has been defined in the text.

**59. L278 Application rate? A rate implies speed of application**

Authors' response: In agronomy and soil science, an application rate refers to the mass of amendment applied per mass or area of soil.

**60. L282 Why the difference between SM biochar and the others? Provide some more detail, mechanisms**

Authors' response: Regarding the change in the amount of nickel bound to organic materials under the influence of biochars, an extensive comparison was made with the literature. But in general, the mechanisms related to it are not completely clear yet and conflicting results have been reported in different studies. Please see the revised manuscript.

**61. L291 - L292 rates? The addition of S0 is not an addition – nothing was added – please rephrase**

Authors' response: The sentence has been rewritten to improve the wording. It now reads: "Furthermore, increasing the Si concentration from $S_0$ to $S_2$ significantly decreased MnOx-Ni by 17.6% (Table 4)."

**62. L294 C+S0 is not a treatment -it is a control with no addition – please rephrase**

Authors' response: The sentence has been rewritten to improve the wording. It now reads: "The interaction effect of treatments showed that the highest and the lowest MnOx-Ni concentrations were found in the untreated control (11.58 mg Ni $kg^{-1}$ soil) and combined SM300+$S_2$ (2.98 mg Ni $kg^{-1}$ soil), respectively (Table 4)."

**63. Table 7 - mg Cd kg-1 h-1??**

Authors' response: We have corrected the unit in Table 7 to Ni as pointed out by the reviewer.

**Without going further this manuscript need a significant re-write before it is in any way acceptable for publication. If the authors revise the manuscript taking into account the above comments and apply the same thoroughness to the remainder of the manuscript I will happily review this again from the beginning.**

---

## Referee Report (RR1)

**Investigating the synergistic potential Si and biochar to immobilize soil Ni in a contaminated calcareous soil after Zea mays L. cultivation**

**ATC1**

**Summary**

I acknowledge the efforts the authors have made to revise and improve the manuscript, for example section 2 is much better. However, significant work is still needed before the manuscript is acceptable for publication. For example in the introduction more information is required in order to provide better context and the relevance/importance of the study. I would appreciate if after making all the corrections that the authors read through the manuscript again carefully, at least twice, as it seems to me, that in a rush to get it re-submitted a plethora of careless mistakes have been made/missed.

**Corrections**

Abstract: there might be no previous study like this one, but why then is it important/relevant? Please say why reducing the bioavailability of Ni is important. How much agricultural soil is contaminated with Ni? The first sentence should start with *"In Iran, X% or a significant percentage of agricultural soils are contaminated with a range of PTEs including Ni, with levels ranging from X to X...."* Then Si is a …..

L19 interactive effects, delete levels,

L20 biochar singular, replace alleviate with "reduce"

L38 replace using with "use of"

L44 replace "it" with Ni

L46-54

- The information on the Shabhazi study should come first to provide the context. *In a study conducted by Shabazi et al….*
- Collected soils from different climates? What does this mean? Climates don't have soils. Climate is typically defined as the long-term patterns of temperature, humidity, wind, precipitation, and other atmospheric conditions in a particular region. Climatic regions? As mentioned in the last review, more background is required. For example something like the following but with references and put more succinctly:

**Land type**

"Iran is a country known for its diverse geography, ranging from deserts to mountainous regions, which affects the availability of land suitable for agriculture. Arable Land: Arable land refers to land suitable for cultivation of crops. In Iran, arable land accounts for approximately 10% to 12% of the total land area. Cultivable Land: This category includes not only arable land but also land suitable for other forms of agriculture such as orchards, vineyards, and pastures. The percentage of cultivable land in Iran is slightly higher than arable land, estimated to be around 16% to 18% of the total land area. Irrigated Land: Iran has a significant reliance on irrigation for agriculture, particularly in arid and semi-arid regions. Irrigated land accounts for approximately 10% to 12% of the total land area. Marginal Land: Some portions of Iran's land are considered marginal for agriculture due to factors such as soil quality, topography, and water availability. The extent of

marginal land varies across regions but can be significant, particularly in arid and mountainous areas. Overall, while Iran has considerable agricultural potential, it also faces challenges such as water scarcity, soil degradation, and limited access to modern farming technologies, which can affect the actual percentage of land available for agriculture and agricultural productivity."

**Climate – use the Köppen classification**

BWh: Hot Desert Climate (Arid):

BSk: Cold Semi-Arid Climate:

Csa: Hot-Summer Mediterranean Climate:

Csb: Warm-Summer Mediterranean Climate:

Dsa: Cold Semi-Arid Climate:

ET: Tundra Climate:

- Characterise the climate, average annual precipitation etc. of your study area
- Please add a summary of the background above before introducing the Shabazi study

All the above information should provide good background to the study being carried out and underscore the importance/relevance of the study.

L47 2 significant figures is enough - 350

L61 mitigate "the toxicity of PTEs….

L63 reduce "the bioavailability of PTEs…

L64 and "the formation of…

L68 Biochar can be used for many things including water filtration. Change to "Biochar *can be used for a number of applications including*….

L81 Increased

L86 organic matter

L89 PTEs don't have chemical fractions. Rephrase, PTEs in other….

L100 the bonding of…

L105 Rewrite the latter half of this sentence after oxides

L107 Explain what light texture means

L111 "..reduce plant PTE (potentially toxic element) uptake… change to plant uptake of PTEs. Again, this PTE word order issues was mentioned previously. Read out loud all the abbreviation in the text in full to ensure correct word order.

L116 change alleviate to reduce

**Section 2 is much better!**

But for section 2.1 I still strongly suggest adding a map of Iran or Southern Iran to show arable regions, where your study area is and how close it is to industrial areas – see below paper as an example

[Figure]

**Figure**

Caption

Study area and sampling points in Kermanshah province

L121 with an auger and placed into what? How was the sample then transported to the lab. How were they stored in the lab before preparation?

L124 Sieve material?

L152 determine**d**

L173 "from" the pots. In what and where was the soil stored before analysis?

L183 millilitre should be singular here. In addition please standardise, millilitre, ml or mL (L 131)?

L195 space between brackets

L197 "between" the individual…

**Section 3**

L214 Low relative to what? Remind the reader that the average for Iran is 350 mg kg-1. Why did you use a soil with such a low Ni content for this experiment?

L236 indicate

L242  3.3 Change to "Biochar analysis using FTIR and SEM"

L255 FTIR "spectra"… delete "of"

L264 SEM "images"

L269-271 You say effect of treatments were significant except for NI-Car fraction, and then you say "where only the main effects were significant". Confusing – please re-write this sentence.

L271-272 for which biochar is this? State the concentrations in addition to the % reduction.

L274 As above, state the concentrations – from xx mkg-1 to xx mg kg-1

L281 For clarity state what is being reduced in the WsEx fraction

L295 delete probably

L296 replace "indicated" with "reflected", replace "by" with "in", replace "higher" with "greater"

L297 replace "reflected" by "indicated". Add "the" between by and lower

L304 Sentence beginning "Due to the…" How does this information from the Ma et al., study relate to this study?

L314 Change to "Si application rates from…

L315 2 at most 3 sig figures is enough. The same goes for Table 4 and Table 3

L318 "mine contaminated soil"? …a soil contaminated by mining activities…

L322 PTEs and word order again. …bioavailability of PTEs in soil…

L321 State what this experiment used as an amendment

L324 "The decrease in the concentration of Ni in the carbonate form with an increase in the Si levels…", to what is this referring? As found in this study? What treatment/s?

L337 "The greatest OM-Ni reduction (18.6%) was due to SM500". Rephrase – see line 314 "… *the greatest reduction in soil Ni content in the OM fraction was found to be in that which underwent the SM500 treatment…*"

L341 Relace "they" with the study

L342 delete "as affected"

L344 "…in the organic matter cadmium fraction…" sounds strange, change to "…. increase in Cd in the OM fraction…"

L348, L349 – see L344

L349 – no fighting going on as far as I am aware. Please replace "conflict" with "contrast"

L350 Change "According to the above-mentioned points, it seems that.." to "These observations point to the fact that… or "These observations indicate…

L354 – see L344

L353-354 For which treatment? Or is this a mean? Please add this info.

Table 4 – where is the Ni mobility factor (%) referred to in the table caption and notes? Is this whole table not concentrations of Ni in mg kg-1?

L354-355 Where has it been shown? References.

L356 Start with "In this study, the interaction….

L357 – see L344

L359 C+S0 is not a treatment it is a control – please rephrase

L360 – see L344

L361 "the" control. Delete "was". Replace "attributed to" with "caused by". ".. the SM300 *treatment". Replace "by" with "at"*

L363 – see L344

L364 .. at "a" lower…

L366 .. at "a" higher…

L367 Delete "the" after increasing

L368 Replace "low" with lower, temperature with temperatures

L367-L369 Where does this information come from? Provide reference or start the sentence with "It is well known….

L371 Change "…the exchangeable and water-soluble Mn concentration…" to ".. the Mn concentration in…"

L373 – see L344

L373-375 The control is not a treatment. Start sentence with "Compared to the control which had the highest concentration of Ni in the MnOx fraction, the greatest interactive effect was….

L376 concentration not concentrations "..the" AFeOx and CFeOx "fractions was…"

L382 – see L344

L384, delete "the" after however. Only the "mean value" of the control and the 2 different applications of Si for the RH500 treatment…

L385 Delete B from SMB

L386-387 – what do you mean by this? By form do you mean "fractions". It seems to me that the addition of Si did affect the Ni concentration in this fraction. E.g. for SM 200, S1 and S2 significantly decreased the concentration of Ni compared to the control. There are other examples for this fraction.

L388 In this study was contaminated or amended soil used?

L390 Ni in the Res fraction and reduced Ni in the other fractions

L392-393 Did the amount of the fractions increase or did the Ni increase in the fractions? Rephrase entire sentence.

L394-396 Ditto

L397 PTE again, rephrase

L401 Ni did not transform, it moved or transferred more into other fractions

L404 replace "use" with "application"

L405 replace was with "is"

L403-405 The RH500 with Si was also effective for this fraction

L407 "application" rates, replace "interactions" with interactive effects

Table 4 and Table 5 captions and text – standardise the addition of Si. You have Si application levels (Tab 5), silicon levels (Tab 4) and "application rates" or just "Si rates" (L407) in the text

L407-408 Change to "With the exception of SM500 (S0) use of biochar, Si application and their interactive effects were all had a statistically significant effect on shoot Ni concentration."

Table 5 (and actually also Table 4): I am not sure of the value of calculating the mean of different treatments. Surely the point is to compare the effects and interactions of the individual treatments? What does taking the mean of the control with no silicon, the control with 250 mg kg Si and the control with 500 mg kg Si bring? Likewise what is the point of taking the mean value of the control with no silicon and 4 different biochars with

no silicon bring? The data of real value in the table are which silicon application level and which biochar reduce the amount of shoot Ni – i.e. SM500 and SM300 S2.

L408-409 32%? For which treatment? Seems to me that SM500 S2 has more than 50% less Ni relative to the control with no silicon and about 50% of the shoot Ni relative to the same biochar with no Silicon.

L409-411 – SM500 (S0) does not have a significantly different shoot Ni concentration compared to the control.

L411 Replace interaction with interactive

L412 C+S0 is not a treatment, it is the control by which all treatments are compared. Rephrase

L413-416 – there is no data shown to confirm this statement and this sentence is a bit random. Perhaps start with something along the lines of that you compared shoot Ni concentrations with Ni concentrations in the soil fractions. From this analysis it was found…. Similarly soil pH.

L419 Change "reduction of shoot Ni concentration of spinach…" to "a reduction in the concentration of Ni in spinach shoots…

L420 Add "of" between application and rice. Delete 2$^{nd}$ application. Was the reduction significant? Say so either way.

L423 PTE – rephrase

L424 What do you mean by "surface adsorption" is a significant factor? Rephrase. Surface adsorption by what?

L425 What do you mean by "altered redox conditions of PTEs", be more specific

L433 Maize not maze, Pb concentration in shoots not Pb-shoot concentration. What are lead-shoots?

L434 Ni concentration in shoots

L438 see L434

L447 Fig. or Figure – standardise throughout according to journal format

Fig. 5 y-axis caption – capital C, y-axis - no need for .00, 2 sig. figures enough. The colour coding for the different treatments could be improved as there is not much difference between the colours of some of the treatments. At least make the control red or some stronger colour to stand out, maybe start the axis at 5 to spread the different points out a bit more.

L453 Add "the" between higher and pyrolysis, delete "the" between reducing and soil

L454 "the" lower…

L455 C+S0 – again, not a treatment – this is the control with nothing added

L464 kinetic

L490 did you contaminate the soil or amend it?

L492 kinetic

L493 metal

L494 "the" treatments. Replace "of all the biochar treatments" with "all 4 biochars"

L497 delete "has"

L498 see L490

L499 interactive, indicate

L510 It looks more like 50% to me – please check

Table 7 caption "the" power… "Ni-polluted"? See L490

L522 see L490

L541 enhances

L543 distribution of Ni between the various soil chemical forms

Conclusions: Please say something about how you think your results arising from a soil amended with Ni compares to an aged Ni-contaminated soil. In other words how representative is your experiment to the real life situation?

---

## Author Response (AR2)

**RESPONSE TO REVIEWER COMMENTS ON MANUSCRIPT: Investigating the synergistic potential Si and biochar to immobilize soil Ni in a contaminated calcareous soil after *Zea mays* L. cultivation Egusphere-2023-2687**

The authors would like to thank anonymous reviewers for their time, invaluable comments and suggestions for substantially improving this manuscript. Please find detailed responses to each comment below.

**ALL CHANGES ARE INDICATED IN GREEN HIGHLIGHT IN THE REVISED MANUSCRIPT**

**ANONYMOUS REFEREE #1**
The manuscript has greatly improved in quality, thank you for accepting the previous suggestions. However, I just have one small issue:

1. **In L. 131 - 136, indeed the methodology of soil incubation with Ni has been developed a little more. However, the incubation conditions: temperature, darkness, duration have not been included. This information must be added.**
   **In addition, I would like a short explanation on how the drying and rewetting cycles contribute to equilibrate Ni with the soil.**

Authors' response: some available information about incubation conditions were added as suggested by the respectable reviewer. Please see the revised manuscript. we developed the drying and rewetting cycles for soil contamination with Ni to simulate real field conditions.

2. **L. 145: "Biochar pH and EC were determined…". Same comment in L. 147 ("Biochar total C, N and H contents were determined…")**

Authors' response: these corrections were done as suggested by the respectable reviewer. Please see the revised manuscript.

3. **Same comment for L. 283 – 284 ("It has been shown that the application of Si to cultivated soils resulted in a reduction of soil organic matter content". Authors' response: Unfortunately, it is currently not possible to measure the amount of soil organic matter. Reviewer response: OK, I understand that it may not be possible to measure the amount of SOM, but then the original sentence "It has been shown that the application of Si to cultivated soils resulted in a reduction of soil organic matter content" should be written in a hypothetical way.**

Authors' response: The sentence given in this section about the reduction of the amount of soil organic matter, as influenced by Si application, is the result of the others (Ma et al. 2021) to confirm the reduction of the amount of Ni in the OM fraction in the present study. The sentence was revised in such a way that the doubts of two referees were completely resolved. Please see the revised manuscript.

**ANONYMOUS REFEREE #2**

**Summary**
I acknowledge the efforts the authors have made to revise and improve the manuscript, for example section 2 is much better. However, significant work is still needed before the manuscript is acceptable for publication. For example in the introduction more information is required in order to provide better context and the relevance/importance of the study. I would appreciate if after making all the corrections

that the authors read through the manuscript again carefully, at least twice, as it seems to me, that in a rush to get it re-submitted a plethora of careless mistakes have been made/missed.

**Corrections**
1. **Abstract: there might be no previous study like this one, but why then is it important/relevant? Please say why reducing the bioavailability of Ni is important. How much agricultural soil is contaminated with Ni? The first sentence should start with "In Iran, X% or a significant percentage of agricultural soils are contaminated with a range of PTEs including Ni, with levels ranging from X to X···." Then Si is a …..**

Authors' response: thanks for the valuable suggestion. We added a brief sentence at the beginning of the abstract to satisfy the opinion of the respectable reviewer. Please see the revised manuscript.

2. **L19 interactive effects, delete levels,**

Authors' response: We have made the corrections within the manuscript as suggested by the respectable reviewer. Please see the revised manuscript.

3. **L20 biochar singular, replace alleviate with "reduce"**

Authors' response: We have made the corrections within the manuscript as suggested by the respectable reviewer. Please see the revised manuscript.

4. **L38 replace using with "use of"**

Authors' response: We have made the corrections within the manuscript as suggested by the respectable reviewer. Please see the revised manuscript.

5. **L44 replace "it" with Ni**

Authors' response: We have made the corrections within the manuscript as suggested by the respectable reviewer. Please see the revised manuscript.

6. **L46-54  The information on the Shabhazi study should come first to provide the context. In a study conducted by Shabazi et al….**
   **Collected soils from different climates? What does this mean? Climates don't have soils. Climate is typically defined as the long-term patterns of temperature, humidity, wind, precipitation, and other atmospheric conditions in a particular region. Climatic regions? As mentioned in the last review, more background is required. For example something like the following but with references and put more succinctly:**
   **Land type**
   **"Iran is a country known for its diverse geography, ranging from deserts to mountainous regions, which affects the availability of land suitable for agriculture. Arable Land: Arable land refers to land suitable for cultivation of crops. In Iran, arable land accounts for approximately 10% to 12% of the total land area. Cultivable Land: This category includes not only arable land but also land suitable for other forms of agriculture such as orchards, vineyards, and pastures. The percentage of cultivable land in Iran is slightly higher than arable land, estimated to be around 16% to 18% of the total land area. Irrigated Land: Iran has a significant reliance on irrigation for agriculture, particularly in arid and semi-arid regions. Irrigated land accounts for approximately 10% to 12% of the total land area. Marginal Land: Some portions of Iran's land are considered marginal for agriculture due to factors such as soil quality, topography, and water availability. The extent of marginal land varies across regions but can be significant, particularly in arid and mountainous areas. Overall, while Iran has considerable agricultural potential, it also faces challenges such as**

water scarcity, soil degradation, and limited access to modern farming technologies, which can affect the actual percentage of land available for agriculture and agricultural productivity."
Climate – use the Köppen classification
BWh: Hot Desert Climate (Arid):
BSk: Cold Semi-Arid Climate:
Csa: Hot-Summer Mediterranean Climate:
Csb: Warm-Summer Mediterranean Climate:
Dsa: Cold Semi-Arid Climate:
ET: Tundra Climate:
 Characterise the climate, average annual precipitation etc. of your study area
 Please add a summary of the background above before introducing the Shabazi
Study
**All the above information should provide good background to the study being carried out and underscore the importance/relevance of the study.**

Authors' response: according to the reviewer suggestion, we changed the sentence to ¨**In a study conducted by Shahbazi et al. (2022)**, **the Ni weighted average concentration of the cultivated lands of Iran in the vicinity of the industrial areas was reported 349.8 mg kg$^{-1}$ soil**¨ for more clarity. Furthermore, we complete the sentence of ¨Shahbazi et al. (2020) collected……¨ based on the referee suggestion and changed it as follows to more clarity: **Shahbazi et al. (2020) collected 711 agricultural soil samples located at different climate zones (extra arid, arid, semi-arid, Mediterranean, semi humid, humid and per-humid based on the de Martonne classification system) of Iran**. As you know, the climate is one of the most important soil-forming factors and the properties of soils at different climate zones are totally different. We also added the climate, mean annual precipitation, soil moisture regime and soil thermal regime of the study area in the materials and methods section (soil sampling). Please see the revised manuscript.

7. **L47 2 significant figures is enough - 350**

Authors' response: We have made the corrections within the manuscript as suggested by the respectable reviewer. Please see the revised manuscript.

8. **L61 mitigate "the toxicity of PTEs….**

Authors' response: We have made the corrections within the manuscript as suggested by the respectable reviewer. Please see the revised manuscript.

9. **L63 reduce "the bioavailability of PTEs…**

Authors' response: We have made the corrections within the manuscript as suggested by the respectable reviewer. Please see the revised manuscript.

10. **L64 and "the formation of…**

Authors' response: We have made the corrections within the manuscript as suggested by the respectable reviewer. Please see the revised manuscript.

11. **L68 Biochar can be used for many things including water filtration. Change to "Biochar can be used for a number of applications including⋯.**

Authors' response: There is no such sentence in the text of our manuscript.

12. **L81 Increased, L86 organic matter, L89 PTEs don't have chemical fractions. Rephrase, PTEs in other…., L100 the bonding of…**

Authors' response: We have made the corrections within the manuscript as suggested by the respectable reviewer. Please see the revised manuscript.

**13. L105 Rewrite the latter half of this sentence after oxides**

Authors' response: we have rewritten it as follows: while the residual and organic matter-bound forms experienced a notable enhancement. Please see the revised manuscript.

**14. L107 Explain what light texture means**
Authors' response: we have added soil textural class of the soil for more clarity. Please see the revised manuscript.

**15. L111 "..reduce plant PTE (potentially toxic element) uptake… change to plant uptake of PTEs. Again, this PTE word order issues was mentioned previously. Read out loud all the abbreviation in the text in full to ensure correct word order.**
Authors' response: We have made the corrections within the manuscript as suggested by the respectable reviewer. Please see the revised manuscript.

**16. L116 change alleviate to reduce**
Authors' response: We have made the corrections within the manuscript as suggested by the respectable reviewer. Please see the revised manuscript.

**17. But for section 2.1 I still strongly suggest adding a map of Iran or Southern Iran to show arable regions, where your study area is and how close it is to industrial areas – see below paper as an example**
Authors' response: We have added a map to the manuscript (2.1) which shows the exact location of soil sampling. Please see the revised manuscript.

**18. L121 with an auger and placed into what? How was the sample then transported to the lab. How were they stored in the lab before preparation?**
Authors' response: After collecting the soil samples from the field by auger, they were mixed completely and after air drying and passing through a 2 mm sieve, the soil sample was placed in a plastic bag and transported to the laboratory. In the laboratory, it was kept in a dark room until treating.

**19. L124 Sieve material?**
Authors' response: thanks for your valuable comment. For better understanding, sieve word replaced with pass. Please see the revised manuscript.

**20. L152 determined**
Authors' response: we have corrected it. Please see the revised manuscript.

**21. L173 "from" the pots. In what and where was the soil stored before analysis?**
Authors' response: we have corrected it. Also, please see the response number 18.

**22. L183 millilitre should be singular here. In addition please standardise, millilitre, ml or mL (L 131)?**
Authors' response: we have corrected them. Please see the revised manuscript.

**23. L195 space between brackets, L197 "between" the individual…**

Authors' response: we have corrected them. Please see the revised manuscript.

**24. L214 Low relative to what? Remind the reader that the average for Iran is 350 mg kg-1. Why did you use a soil with such a low Ni content for this experiment?**

Authors' response: Here, the respected referee should be reminded that according to the study of Shahbazi et al. (2020), the average Ni in the soils of different regions of Iran is 68 mg kg$^{-1}$ of soil. The average of 350 mg kg$^{-1}$ is related to the soils in the vicinity of industrial areas (Shehbazi et al. 2022) (see the introduction section). In general, according to Iran's environmental standard, if the soil contains more than 100 mg Ni kg$^{-1}$ soil, it is considered contaminated. For this reason, we chose the 300 mg Ni kg$^{-1}$ soil for soil contaminating in this study.

**25. L236 indicate, L242 3.3 Change to "Biochar analysis using FTIR and SEM", L255 FTIR "spectra"… delete "of", L264 SEM "images",**

Authors' response: we have corrected them. Please see the revised manuscript.

**26. L269-271 You say effect of treatments were significant except for NI-Car fraction, and then you say "where only the main effects were significant". Confusing – please re-write this sentence.**

Authors' response: thanks for your valuable comment. We have rewritten the sentence for better understanding. please see the revised manuscript.

27. **L271-272 for which biochar is this? State the concentrations in addition to the % reduction.**

Authors' response: Here, we have discussed on the main effects of Si levels on reduction of WsEx fraction. We also added the concentrations. Please see the revised manuscript.

**28. L274 As above, state the concentrations – from xx mkg-1 to xx mg kg-1**

Authors' response: We have added the concentration changes. Please see the revised manuscript.

29. **L281 For clarity state what is being reduced in the WsEx fraction**

Authors' response: for better clarity, we have added Ni to WsEx (Ni-WsEx). Please see the revised manuscript.

**30. L295 delete probably, L296 replace "indicated" with "reflected", replace "by" with "in", replace "higher" with "greater", L297 replace "reflected" by "indicated". Add "the" between by and lower**

Authors' response: we have corrected them. Please see the revised manuscript.

**31. L304 Sentence beginning "Due to the…" How does this information from the Ma et al., study relate to this study?**

Authors' response: In this research, the source of Si was sodium metasilicate. Because the application of Si levels in the present study increased soil pH, the presence of this sentence is to confirm the content and how sodium metasilicate increases soil pH.

**32. L314 Change to "Si application rates from…,, L315 2 at most 3 sig figures is enough. The same goes for Table 4 and Table 3, L318 "mine contaminated soil"? …a soil contaminated by mining activities…, L322 PTEs and word order again. …bioavailability of PTEs in soil…,,**

Authors' response: we have corrected them. Please see the revised manuscript.

**33. L321 State what this experiment used as an amendment**

Authors' response: the type of amendment material was added. Please see the revised manuscript.

**34. L324 "The decrease in the concentration of Ni in the carbonate form with an increase in the Si levels…", to what is this referring? As found in this study? What treatment/s?**

Authors' response: thanks for your valuable suggestion. we moved this sentence to the desired location. Please see the revised manuscript.

**35. L337 "The greatest OM-Ni reduction (18.6%) was due to SM500". Rephrase – see line 314 "…the greatest reduction in soil Ni content in the OM fraction was found to be in that which underwent the SM500 treatment…"**

Authors' response: we have corrected the sentence as suggested by the respectable reviewer. Please see the revised manuscript.

**36. L341 Relace "they" with the study,**

Authors' response: It seems that it is more correct to use the word of ˝they˝ here.

**37. L342 delete "as affected", L344 "…in the organic matter cadmium fraction…" sounds strange, change to "…. Increase in Cd in the OM fraction…",**

Authors' response: we have corrected them. Please see the revised manuscript.

38. **L348, L349 – see L344, L349 – no fighting going on as far as I am aware. Please replace "conflict" with "contrast", L350 Change "According to the above-mentioned points, it seems that.." to "These observations point to the fact that… or "These observations indicate…, L354 – see L344,**

Authors' response: we have corrected them. Please see the revised manuscript.

**39. L353-354 For which treatment? Or is this a mean? Please add this info.**

Authors' response: in this sentence, the treatment is clear.  By application **the Si levels**……

**40. Table 4 – where is the Ni mobility factor (%) referred to in the table caption and notes? Is this whole table not concentrations of Ni in mg kg-1?**

Authors' response: thanks for your accuracy. We have deleted the Ni mobility factor from the topic of Table 4. Please see the revised manuscript.

**41. L354-355 Where has it been shown? References.**

Authors' response: The sentence was revised in such a way that the doubts created for both referees in this part are resolved. Please see the revised manuscripts.

**42. L356 Start with "In this study, the interaction…., L357 – see L344, L359 C+S0 is not a treatment it is a control – please rephrase, L360 – see L344, L361 "the" control. Delete "was". Replace "attributed to" with "caused by". ".. the SM300 treatment". Replace "by" with "at"**

Authors' response: we have corrected them. Please see the revised manuscript.

**43. L363 – see L344, L364 .. at "a" lower…,, L366 .. at "a" higher…,, L367 Delete "the" after increasing,, L368 Replace "low" with lower, temperature with temperatures,,**

Authors' response: we have corrected them. Please see the revised manuscript.

**44. L367-L369 Where does this information come from? Provide reference or start the sentence with "It is well known….**

Authors' response: thanks for your valuable suggestion. we have added the Phrase of "It is well known…. In the beginning of sentence. Please see the revised manuscript.

**45. L371 Change "…the exchangeable and water-soluble Mn concentration…" to ".. the Mn concentration in…" L373 – see L344, L373-375 The control is not a treatment. Start sentence with "Compared to the control which had the highest concentration of Ni in the MnOx fraction, the greatest interactive effect was…. L376 concentration not concentrations "..the" AFeOx and CFeOx "fractions was…" L382 – see L344,**

Authors' response: we have corrected them. Please see the revised manuscript.

**46. L384, delete "the" after however. Only the "mean value" of the control and the 2 different applications of Si for the RH500 treatment…**

Authors' response: Here, there is no discussion of silicon application levels. Hear, the main effects of biochars application on changes the concentration of soil Ni in the form of AFeOx is given.

**47. L385 Delete B from SMB**

Authors' response: we have corrected it. Please see the revised manuscript.

**48. L386-387 – what do you mean by this? By form do you mean "fractions". It seems to me that the addition of Si did affect the Ni concentration in this fraction. E.g. for SM 200, S1 and S2 significantly decreased the concentration of Ni compared to the control. There are other examples for this fraction.**

Authors' response: Here, the main effects of the Si levels on the changes of the soil Ni Res fraction is considered. The sentence was modified for better clarity. Please see the revised manuscript.

**49. L388 In this study was contaminated or amended soil used?**

Authors' response: as clearly described in the Materials and Methods section, in this study, we contaminated the calcareous soil with Ni in the level of 300 mg Ni kg$^{-1}$ soil (it is marked with "Ni-contaminated soil" within the text of the manuscript) and then, we apply the Si and biochar as amendment materials for Ni immobilizing process.

**50. L390 Ni in the Res fraction and reduced Ni in the other fractions**

Authors' response: we have corrected it. Please see the revised manuscript.

51. **L392-393 Did the amount of the fractions increase or did the Ni increase in the fractions? Rephrase entire sentence. L394-396 Ditto**

Authors' response: we have corrected the sentence in order to more clarify as suggested by the respectable reviewer. Please see the revised manuscript.

52. **L397 PTE again, rephrase, L401 Ni did not transform, it moved or transferred more into other fractions**,

Authors' response: we have corrected it. We replaced the word of ˝transformation˝ by ˝transfer˝. Please see the revised manuscript.

53. **L404 replace "use" with "application", L405 replace was with "is"**

Authors' response: we have corrected them. Please see the revised manuscript.

54. **L403-405 The RH500 with Si was also effective for this fraction**

Authors' response: its true. But for the SM treatments were more effective than others. We have added the word of ˝more˝ before ˝evident˝ in the sentence for better clarity.

55. **L407 "application" rates, replace "interactions" with interactive effects**

Authors' response:  we have corrected them. Please see the revised manuscript.

56. **Table 4 and Table 5 captions and text – standardize the addition of Si. You have Si application levels (Tab 5), silicon levels (Tab 4) and "application rates" or just "Si rates" (L407) in the text**

Authors' response: as suggested by the respectable reviewer, we standardized it as Si application levels. Please see the revised manuscript.

57. **L407-408 Change to "With the exception of SM500 (S0) use of biochar, Si application and their interactive effects were all had a statistically significant effect on shoot Ni concentration."**

Authors' response: Here, it is meant to express the significance of the main effects of biochar treatments and Si levels and their interactions on the Ni shoot content base on the ANOVA.

58.  **Table 5 (and actually also Table 4): I am not sure of the value of calculating the mean of different treatments. Surely the point is to compare the effects and interactions of the individual treatments? What does taking the mean of the control with no silicon, the control with 250 mg kg Si and the control with 500 mg kg Si bring? Likewise what is the point of taking the mean value of the control with no silicon and 4 different biochars with no silicon bring? The data of real value in the table are which silicon application level and which biochar reduce the amount of shoot Ni – i.e. SM500 and SM300 S2.**

Authors' response: When the data are analyzed by statistical software, the significance of the main effects of the treatments (levels of Si and biochars each alone) and their interactions are determined. The purpose of expressing the main effects of treatments, for example, the effect of biochar application, is to determine the best biochar treatment, for instance, in reducing the Ni concentration of shoots, without considering the effect of Si application. The purpose of expressing interactive effects is to compare the effect of all the combined treatments of biochar and Si (15 treatments) and determine the best combined treatment, for example, in reducing the concentration of Ni in the shoot.

**59. L408-409 32%? For which treatment? Seems to me that SM500 S2 has more than 50% less Ni relative to the control with no silicon and about 50% of the shoot Ni relative to the same biochar with no Silicon. L409-411 – SM500 (S0) does not have a significantly different shoot Ni concentration compared to the control**

Authors' response: please see the above response. we fist indicated the main effects of Si levels and biochar, separately and in following, the interaction effects are given. 32% decrease in shoot Ni concentration is associated with the main effects of Si application levels.

**60. L411 Replace interaction with interactive, L412 C+S0 is not a treatment, it is the control by which all treatments are compared. Rephrase,**

Authors' response: we have corrected them. Please see the revised manuscript.

61. **L413-416 – there is no data shown to confirm this statement and this sentence is a bit random. Perhaps start with something along the lines of that you compared shoot Ni concentrations with Ni concentrations in the soil fractions. From this analysis it was found…. Similarly soil pH**.

Authors' response: In this manuscript, we have presented all the data related to the chemical fractions of Ni in the soil, the concentration of Ni in the plant and the soil pH under the influence of the application of Si and biochar levels. In this part, we have brought the results of Pearson correlation between these parameters (correlation coefficient and probability level) which has been done by SPSS software and conclusions have been introduced according to it. It seems that, the content is very well-founded and logical.

**62. L419 Change "reduction of shoot Ni concentration of spinach…" to "a reduction in the concentration of Ni in spinach shoots…L420 Add "of" between application and rice. Delete 2nd application. Was the reduction significant? Say so either way. L423 PTE – rephrase**

Authors' response: we have corrected them. Please see the revised manuscript.

**63. L424 What do you mean by "surface adsorption" is a significant factor? Rephrase. Surface adsorption by what? L425 What do you mean by "altered redox conditions of PTEs", be more specific. L433 Maize not maze, Pb concentration in shoots not Pb-shoot concentration. What are lead-shoots? L434 Ni concentration in shoots, L438 see L434**

Authors' response: we have corrected them. Please see the revised manuscript.

**64. L447 Fig. or Figure – standardize throughout according to journal format**

Authors' response: When this word is used in the text, it is written in full, and when it is in parentheses, it is abbreviated. Abbreviations are also given for figure captions. Please see the revised manuscript.

**65. Fig. 5 y-axis caption – capital C, y-axis - no need for .00, 2 sig. figures enough. The colour coding for the different treatments could be improved as there is not much difference between the colours of some of the treatments. At least make the control red or some stronger colour to stand out, maybe start the axis at 5 to spread the different points out a bit more.**

Authors' response: we have done the corrections for the Figure 6 as suggested by the respectable reviewer. Please see the revised manuscript.

**66. L453 Add "the" between higher and pyrolysis, delete "the" between reducing and soil. L454 "the" lower… L455 C+S0 – again, not a treatment – this is the control with nothing added, L464 kinetic,**

Authors' response: we have corrected them. Please see the revised manuscript.

**67. L490 did you contaminate the soil or amend it?**

Authors' response: please refer to the response number 49.

**68. L492 kinetic, L493 metal, L494 "the" treatments. Replace "of all the biochar treatments" with "all 4 biochars" , L497 delete "has", L498 see L490, L499 interactive, indicate,**

Authors' response: we have corrected them. Please see the revised manuscript.

**69. L510 It looks more like 50% to me – please check**

Authors' response: it was checked again. It was true.

**70. Table 7 caption "the" power… "Ni-polluted"? See L490, L522 see L490**

Authors' response: the correction was done.  Also, please refer to the response number 49.

71. **L541 enhances,**

Authors' response: the correction was done. Please see the revised manuscript.

**72. L543 distribution of Ni between the various soil chemical forms**

Authors' response: This phrase (soil chemical forms) has no correct meaning. We changed to ¨the distribution of Ni chemical fractions in soil¨ for better clarity. Please see the revised manuscript.

**73. Conclusions: Please say something about how you think your results arising from a soil amended with Ni compares to an aged Ni-contaminated soil. In other words how representative is your experiment to the real life situation?**

Authors' response: In this experiment, we have not worked on aged Ni- contaminated soils and here, we can only add it as a suggestion for future experiments. Please see the revised manuscript.

---

## Author Response (AR3)

**Response to editor comments on Revision 2 of egusphere-2023-2687**

The authors would like to thank the editor and reviewers for their time and effort for reviewing and improving this manuscript. We apologize for misunderstanding some of the previous comments.

**Reviewer 1**

1. **Page 1: Please, make sure that reviewer's comments made on some areas of the text are applied along the whole manuscript including abstract, introduction, methods, results, discussion and conclusions. As highlighted several times by the reviewers "Ni/Cd/PTE do not have chemical fractions". All these expressions need to be rewritten to "Ni/Cd/PTE in XXXX fraction/s..."**

Authors' response: This correction has been implemented throughout the manuscript as requested by the reviewers.

2. **Page 2, lines 71-73: This sentence is not accurate. Biochar can be used as soil amendment. It is not necessarily a soil amendment. This needs rewriting, including addressing the comment of the reviewer: Biochar can be used for many things including water filtration. Change to "Biochar can be used for a number of applications including…**

Authors' response: The sentence has been amended as follows: "Biochar can be used for a number of applications including as a soil amendment that sequesters soil carbon (C) and for stabilization of PTEs in polluted sites (El-Naggar et al., 2018).

3. **Page 3, line 109: enhancement or increase?**

Authors' response: We have changed the word to increase as suggested by the reviewer.

4. **Page 3, line 129-134: The authors have not addressed the reviewer comment "L121 with an auger and placed into what? How was the sample then transported to the lab. How were they stored in the lab before preparation?"**

Authors' response: the missing information has been added as follows: "The composite soil sample was placed in polyethylene bags in the field and then transported to laboratory where it was immediately air-dried, passed through a 2 mm sieve and then stored at room temperature until the physicochemical analysis was performed."

5. **Page 4, lines 143-144: Review punctuation. That sentence does not need to be in parentheses. A period is needed between sentences. Also, this sentence does not fully answer the reviewer comment "I would like a short explanation on how the drying and rewetting cycles contribute to equilibrate Ni with the soil".**

Authors' response: The parentheses have been removed, and the requested information has been added as follows:" The repeated wetting and drying cycles were performed to simulate field processes."

6. **Page 5, lines 184-187: The following reviewer's comment has not been addressed " "from" the pots. In what and where was the soil stored before analysis?"**

Authors' response: The requested information has been added as follows:" After separating the roots, the soil from the pots was immediately air-dried, and then passed through a 2 mm sieve and stored in

labelled polyethylene bags at room temperature, to be subsequently utilized for performing Ni sequential extraction and release kinetics"

**7. Page 6, line 195: Please maintain coherence along the text. If you say ml, use ml all along. Also, I do not understand why it is used fifty-milliliter here, instead of 50 ml.**

Authors' response: The correction has been made as suggested by the reviewer.

**8. Page 6, lines 213-223: Authors have not addressed reviewer's concerns about "low relative to what" comments. There are lots of "low" properties described in this paragraph. Low should be stated compared with some other situations.**

Authors' response: The respectable reviewer has mentioned a good point. We compared the soil properties values in this experiment to exact values in the cited literatures for other situations (calcareous soils of Iran located at different regions). Please see the revised manuscript.

**9. Page 10, lines 289-292: This sentence is hard to understand. Please rewrite.**

Authors' response: This sentence has been rewritten to improve clarity as follows: "The interaction of treatments (biochars and Si levels) had a statistically significant effect (P<0.01) on Ni concentration in all the soil chemical fractions, except for the Car fraction. Whereas the main effects of individual treatments (biochars and Si levels) on Ni concentration in all the soil chemical fractions were significant."

**10. Page 10: I consider this is not a satisfactory solution to reviewer's comment " For clarity state what is being reduced in the WsEx fraction". What it is reduced in Ni in WsEx fraction, not the fraction itself. This "solution" to reviewer's comments is a clear example of other comment by the same reviewer " I would appreciate if after making all the corrections that the authors read through the manuscript again carefully, at least twice, as it seems to me, that in a rush to get it re-submitted a plethora of careless mistakes have been made/missed."**

Authors' response: All such instances have been corrected so that it now reads "Ni concentration in the WsEx fraction. This correction has been made throughout the manuscript to improve clarity.

**11. Page 11, line 321: The comment of the reviewer, which has been answered in the separated document, has not been addressed here. "Sentence beginning "Due to the…" How does this information from the Ma et al., study relate to this study?"**

Authors' response: We agree with the reviewer that this statement does not directly relate to the previous sentences, and thus we have rewritten and moved the sentences to Page 11, lines 312-315, where it directly relates to the discussion on the synergistic decrease in Ni content in the WsEx fraction in the SM500+S2 treatment.

**12. Page 12, Table 4: There are 4 significant figures in those numbers in yellow, while there are three in the green ones, and the rest of the table. Please check for consistency in the number of significant digits along the manuscript.**

Authors' response: The number of significant digits has been corrected in Table 4, and throughout the manuscript as suggested by the reviewer.

**13. Page 12, line 338-342: I do not think that moving the sentence to this part of the manuscript fully addresses reviewer's comment "The decrease in the concentration of Ni in the carbonate form with an increase in the Si levels…", to what is this referring? As found in this study? What treatment/s?"**

Authors' response: We have added a sentence to help explain the discussion of the data based on the statistical analysis (2-way ANOVA) of the data: "As there was no significant interaction effect between biochar type and Si levels on the Ni concentration in the Car fraction, only the significant individual main effects of biochar and Si levels are shown in Fig. 5." This explains why only the main effect of Si level on Ni content in Car fraction is discussed in lines 339-341, as also clearly shown in Figure 5. Presenting the results in this manner is consistent with the results of the factorial 2-way ANOVA.

**14. Page 13, line 364: Use capital letters after a period.**

Authors' response: This correction has been made as pointed out by the reviewer.

**15. Page 14, line 379 : indicate that...**

Authors' response: This correction has been made as pointed out by the reviewer.

**16. Page 14, lines 382-386: These two sentences (from lines 373 to 376) make no sense and they do not fully answer the reviewer question "Same comment for L. 283 – 284 ("It has been shown that the application of Si to cultivated soils resulted in a reduction of soil organic matter content". Authors' response: Unfortunately, it is currently not possible to measure the amount of soil organic matter. Reviewer response: OK, I understand that it may not be possible to measure the amount of SOM, but then the original sentence "It has been shown that the application of Si to cultivated soils resulted in a reduction of soil organic matter content" should be written in a hypothetical way"**

Authors' response: These sentences have been rewritten to improve the language and intended meaning, as follows:" Changing the Si levels from S0 to S2 reduced the Ni concentration in the OM fraction by 16.8% (Table 4). Ma et al. (2021) reported that the application of Si to cultivated soils significantly reduced soil organic matter content, which could explain why the Ni concentration in OM fraction was reduced in the present study. They indicated that Si facilitates the decomposition of organic matter by enhancing soil pH."

**17. Page 14, line 398: known that...**

Authors' response: This correction has been made as pointed out by the reviewer.

**18. Page 14, line 405: a comma is needed before which.**

Authors' response: This correction has been made as pointed out by the reviewer.

**19. Page 14, line 420: This "main effects" is unclear. Main on statistical analysis?? based on what?**

Authors' response: Main effects refers to the results of the 2-way ANOVA, where interaction effects as well as main effects of the two factors (i.e. biochar type and Si level) were assessed. Actually, main effects of Si application levels are including averages amongst biochar treatments, at each Si application level, as shown in the table by ''mean''. The sentence was change to "The main (mean) effects of Si application showed that increasing the Si levels from $S_0$ to $S_2$ had no statistically significant effect on the Ni content in the Res fraction" to further clarity.

**20. Page 15, line 437: remove particularly, and quantify the "more"**

Authors' response: The word particularly has been removed. The relative decrease in Ni content in WsEx fraction have been added in parentheses to quantify the synergistic effect.

21. **Page 15: This concern has not been addressed "Seems to me that SM500 S2 has more than 50% less Ni relative to the control with no silicon and about 50% of the shoot Ni relative to the same biochar with no Silicon." If 32% is an average amongst treatments (instead of segregated by biochar addition), this needs to be clarified.**

Authors' response: This sentence shows the main effects of Si application on the change of Ni concentration in the shoots (without considering the effects of biochar application). In other words, the averages amongst biochar treatments, at each level of Si application, have been compared statistically. For better clarity, In Table 5, the main effects of Si levels (averages amongst biochar treatments) are shown in the right part of the table in bold, vertical and with capital significant letters, while the main effects of biochars (averages amongst Si treatments, at each biochar type) are shown in the bottom part and in bold, horizontal and with capital significant letters. The interactive effects (15 treatments) have been also shown in the center part with small significant letters. In order to further clarity, we changed the sentence to:" The main (mean) effects of Si application levels showed that changing the Si levels from $S_0$ to $S_2$ resulted in 32% decrease in the Ni concentration in shoots from 8.56 mg Ni $kg^{-1}$ dry matter (DM) to 5.82 mg Ni $kg^{-1}$ DM (Table 5)". Also, for indicating the interactive effects, the sentence was changed to:" The interactive effects of treatments indicated that the lowest Ni concentration in shoots was observed in the combined treatment of **SM500+S$_2$** (4.45 mg Ni $kg^{-1}$ DM), which showed a **57.2% decrease** compared to the control (CS$_0$: without Si and biochar addition) (10.4 mg Ni $kg^{-1}$ DM) (Table 5)".

22. **Page 15, lines 451: To address reviewer's comment (#61), I would suggest to add the full suite of these results (Pearson coeficients of all variables, as stated in the methods) as supplementary material. Comment was: "L413-416 – there is no data shown to confirm this statement and this sentence is a bit random. Perhaps start with something along the lines of that you compared shoot Ni concentrations with Ni concentrations in the soil fractions. From this analysis it was found…. Similarly soil pH"**

Authors' response: The results of the Pearson correlation were added to the supplementary information as suggested by the reviewer.

23. **Page 16, Table 5: Add this is mean.**

Authors' response: The word mean has been added to Table 5 as pointed out by the reviewer.

24. **Page 20: I do not consider that this sentence is a satisfactory solution to the reviewer's comment "Please say something about how you think your results arising from a soil amended with Ni compares to an aged Ni-contaminated soil. In other words how representative is your experiment to the real life situation?"**

Authors' response: A statement has been added to emphasize that the present study deals with recently contaminated soils. It is difficult to say how the results would differ from aged Ni-contaminated soils, as even aged field sites contain PTEs in the more exchangeable and soluble fractions, which would react with the amendments in this study.

Dear editor

The manuscript was carefully edited by one of the authors whose first language is English to improve the writing quality as suggested by the respectable editor. In the two rounds of review, we have used our best efforts to correct and improve the quality of the manuscript based on the reviewers and editor comments in order to publish it in Soil Journal. In the recent revision, an attempt was made to read the manuscript several times and eliminate possible writing errors. We apologize again for misunderstanding some of the previous comments. We look forward to your positive response.

**Corresponding author**

**Dr. Hamid Reza Boostani**

**Shiraz University, Iran.**

---

## Author Response (AR4)

**Response to editor comments on Revision 3 of egusphere-2023-2687**

**The authors would like to thank the topic editor for time and effort for reviewing and improving this manuscript.**

**Topic editor**

Dear authors,

Thank you for the effort to amend the manuscript. I consider that the manuscript has been significantly improved and is appropriate for publication.
 Please, submit a new version of the manuscript that includes the word "Mean" in the correct place in Table 7 and in the first table of the supplementary material.
 Best wishes,

Maria

**Authors' response**: These corrections have been implemented as requested by the topic editor.

Notification to the authors by Sarah Buchmann

 I just noticed that your figure 1 contains a map/aerial. To clarify whether a copyright statement or a credit must be given in the map itself or in the caption, we differentiate between (a) maps entirely created by you, (b) maps created by you but based on layers reused from other originators, or (c) maps simply reused from other originators. An example for (a) is a digital elevation model (DEM) purely based on measurement points collected by you and derived by using a software product. If you use an existing map layer from another originator as a basis for significantly enriching the map with your own content, this would be an example for case (b). Case (c) could be a pure reproduction of Google Maps where your own contribution is rather small (e.g. a city map where you only added a few marks for your study locations). If the map was entirely created by you (case a), there is no need to change the caption or map. Please simply inform us. To the contrary, if your map follows cases (b) or (c), please let us know whether the map is distributed under public domain. If yes, please do not include a copyright statement (copyright is waived) but consider adding a credit to the map or caption. However, if your map follows cases (b) or (c) and is not distributed under public domain, please include at least a credit or even a copyright statement (e.g. © Google Maps), if this is required by the map provider, in the map itself or in the caption.

**Authors' response**: thanks for your information. The necessary information was added within the map.